# Screen-SBERT: Embedding Functional Semantics of GUI Screens to Support GUI Agents

## Abstract

Recent GUI agent studies show that augmenting LLM prompts with app-related knowledge constructed during a pre-exploration phase can effectively improve task success rates. However, retrieving relevant knowledge from the knowledge base remains a key challenge. Existing approaches often rely on structured metadata such as view hierarchies, which are frequently unavailable or outdated, thereby limiting their generalizability. Purely vision-based methods have emerged to address this issue, yet they typically compare GUI elements only by visual appearance, leading to mismatches between functionally different elements. We consider a two-stage retrieval framework, where the first stage retrieves screenshots sharing the same functional semantics, followed by fine-grained element-level retrieval. This paper focuses on the first stage by proposing Screen-SBERT, a purely vision-based method for embedding the functional semantics of GUI screenshots and retrieving functionally equivalent ones within the same mobile app. Experimental results on real-world mobile apps show that Screen-SBERT is more effective than several baselines for retrieving functionally equivalent screenshots. As a result, (1) we formally define the concepts of functional equivalence and functional page class; (2) design a contrastive learning-based embedding framework; and (3) conduct ablation studies that provide insights for future model design.

## 1 Introduction

As the potential action decision capabilities of large language models (LLMs) come into the spotlight, GUI agents, which leverage LLMs to autonomously perceive, interpret, and interact with Graphical User Interfaces, are emerging as promising systems for automating complex tasks in mobile apps. Recent GUI agent studies show that augmenting LLM prompts with app-related knowledge constructed during a pre-exploration is an effective approach. In this paradigm, agents first explore an app to organize GUI elements into custom representation formats stored in a knowledge base. During task execution, they retrieve relevant information and incorporate the functional context of GUI elements into the action decision prompt. This enables LLMs to interpret GUI elements not seen during pretraining and in turn significantly improve task success rates.

However, retrieving appropriate information from the knowledge base is a highly challenging task. Many existing systems (Zhang et al., 2025; Li et al., 2024; Wen et al., 2024) rely on structured metadata such as view hierarchy (VH) files or app source code, but such reliance limits platform generalizability (Feiz et al., 2022; Wang et al., 2024; Lu et al., 2024). Moreover, in real-world scenarios, this metadata is frequently inaccessible, incomplete, or outdated (Fu et al., 2024), making it impractical as a universal solution.

Some approaches, such as Hu et al. (2024); Xie et al. (2025), embed GUI elements with visual encoders like CLIP (Radford et al., 2021) to retrieve knowledge, thereby reducing reliance on structured metadata. Nevertheless, these methods compare a given element against all entries in the knowledge base using only its visual appearance, which can lead to incorrect matches with elements serving different functions despite visual similarity. To address this, we are researching a two-stage retrieval system: the first stage performs retrieval at the screenshot level, and the second stage performs fine-grained element-level retrieval based on the functional semantics of the entire screenshot. This paper aims to embed the functional semantics of screenshots and retrieve functionally equiva-

Figure 1: Examples of functionally equivalent and different pages. Screenshots connected by blue lines belong to the same functional class, while others represent different functional classes.

lent ones within the same mobile app, representing a key component of our ongoing project of the two-stage retrieval system.

Meanwhile, retrieving screenshots that share the same functional semantics is challenging for the following reasons: (1) In social media and shopping apps, the content and products displayed within functionally equivalent screenshots change frequently; (2) The aforementioned limitations necessitate a purely vision-based approach to determine functional equivalence between screenshots without relying on metadata; (3) Methods to address this problem have not yet been sufficiently explored, and, in particular, the problem definition itself has not yet been formally established; (4) A large-scale public dataset dedicated to this task is absent, which further restricts feasible approaches to few-shot learning—a challenge that remains nontrivial.

As an early exploratory study tackling the problem of retrieving screenshots with the same functional semantics using a purely vision-based approach, this work makes the following contributions: (1) Define the concepts of "functional equivalence" and "functional page class" to more clearly articulate the problem; (2) Propose the Screen-SBERT framework, which embed the functional semantics of GUI screenshots in a purely vision-based manner; (3) Introduce a contrastive learning strategy that enables effective few-shot learning and achieves strong performance on small datasets; (4) Reinterpret and discuss the approaches of previous studies in the process of designing the framework, and derive several insights through ablation studies to guide future model design.

## 2 RELATED WORKS ON GUI SCREENSHOT EMBEDDING

Representative studies on GUI screenshot embedding include Screen2Vec (Li et al., 2021) and Screen2Words (Wang et al., 2021), both of which embed the overall context of a screenshot by leveraging GUI element modalities extracted from the view hierarchy (VH). These works pioneered context-aware embeddings of GUI elements to capture the overall semantics of a screen. However, their reliance on VH imposes the limitations mentioned in Section 1. To address this, more recent approaches have explored purely vision-based methods, employing object detection models such as Faster R-CNN (Ren et al., 2015) to detect GUI elements and extract their coordinates.

Feiz et al. (2022); Wu et al. (2023); Fu et al. (2024) are particularly noteworthy studies that draw inspiration from natural language processing (NLP), where individual words are treated as tokens for sentence embedding. In this paradigm, the entire screenshot is regarded as a sentence, and each GUI element is treated as a word token. This analogy enables the application of various NLP training techniques to the task of screenshot embedding. For instance, Wu et al. (2023) and Fu et al. (2024) adopt a pretraining strategy analogous to BERT's masked language modeling (MLM) (Devlin et al., 2019), where some GUI elements are masked and the model is trained to reconstruct them.

However, some studies in NLP have shown that model performance tends to degrade when pretraining data is limited (Kaplan et al., 2020; Zhang et al., 2020; Pérez-Mayos et al., 2021). This limitation is particularly problematic for our task, as collecting large-scale, task-specific datasets is challenging, and widely used screenshot datasets—such as Rico (Deka et al., 2017)—are ill-suited for distinguishing between pages within the same app. In our case, constraints on time and manpower limited us to a small dataset, under which MLM-based pretraining yielded limited performance.

Similar to our work, Feiz et al. (2022) also addresses the problem of identifying screenshot level similarity within the same app. Their model is trained with binary cross-entropy (BCE) loss using annotations that indicate whether the two screens in a given pair are the same. It adopts a cross-encoder architecture, inspired by BERT, in which GUI elements from both screenshots are jointly fed into a single transformer encoder, and similarity is predicted from the [CLS] token output.

While the cross-encoder is effective for precisely determining similarity between two inputs, it is inefficient for retrieval tasks due to the need for pairwise evaluations with every candidate. Sentence-BERT (SBERT) (Reimers & Gurevych, 2019) addressed this limitation by introducing a bi-encoder architecture that embeds each input independently using the same encoder and computes cosine similarity between embeddings. This approach is significantly more cost-efficient for retrieval. For instance, to retrieve the most similar item among 400 candidates, the cross-encoder requires 400 forward passes for all query-candidate pairs. In contrast, the SBERT-based approach allows all candidates to be pre-embedded in advance, requiring only a single forward pass for the query. Inspired by this, we propose Screen-SBERT, which applies a similar approach to GUI screenshot embedding.

Although not specifically designed for mobile GUI screenshots, recent works such as Ma et al. (2024); Jiang et al. (2025); Liu et al. (2025); Yu et al. (2025) have attempted to embed multimodal document screenshots by fine-tuning the vision encoders of vision–language models (VLMs) using contrastive learning. In this paradigm, a multimodal document screenshot is embedded directly without a separate parsing stage. However, achieving sufficient representational capacity with this approach requires a large VLM, and training such models with contrastive learning typically necessitates multiple GPUs. In environments where multiple GPUs are unavailable—particularly when training must be done on a single GPU—the most feasible option is to employ a smaller VLM such as CLIP-Base. Nevertheless, smaller VLMs often fail to deliver satisfactory performance.

Therefore, in computationally constrained environments, embedding GUI screenshots requires the development of an even smaller model than standard VLMs. For this purpose, introducing a separate GUI parsing stage—and subsequently treating each GUI element as a word token as explored in prior studies—may be an effective approach.

## 3  DEFINITIONS

**Functional Equivalence:** We define two screens as functionally equivalent if the user can access the same functionalities through both, irrespective of the specific content displayed. Conversely, two screens are considered functionally different if they provide different functionalities, even when their visual appearance is similar. Figure 1 illustrates examples corresponding to our definition, and additional examples are provided in Appendix A.

**Functional Page Class:** We define a "functional page" as a class and a "screenshot" as an instance. Specifically, a page class refers to a group of screenshots that are functionally equivalent. These classes are annotated manually based on human intuition. Formally, we represent a page class as:

$$c = \{s_1, s_2, \ldots, s_k\} \tag{1}$$

where $c$ denotes a page class and $s_j$ is the $j$-th screenshot in that class.

## 4  DATASET

Based on these definitions, we manually constructed a dataset consisting of three social media apps—Instagram, Facebook, and X—and three shopping apps—Amazon Shopping, Coupang, and Temu. Each app contains hundreds of screenshots categorized into dozens of page classes. Formally, an app is represented as:

$$\text{App} := \{c_1, c_2, \ldots, c_N\} \quad \text{where } c_i = \{s_1^{(i)}, s_2^{(i)}, \ldots, s_{k_i}^{(i)}\} \tag{2}$$

Here, the number of screenshots $k_i$ varies depending on the frequency and accessibility of the page. For pages frequently visited in typical usage flows, each class contains dozens of samples. In contrast, pages that are rarely visited in typical usage flows consist of only a few samples. In extreme cases, some classes are singletons, containing only one screenshot with no functionally equivalent

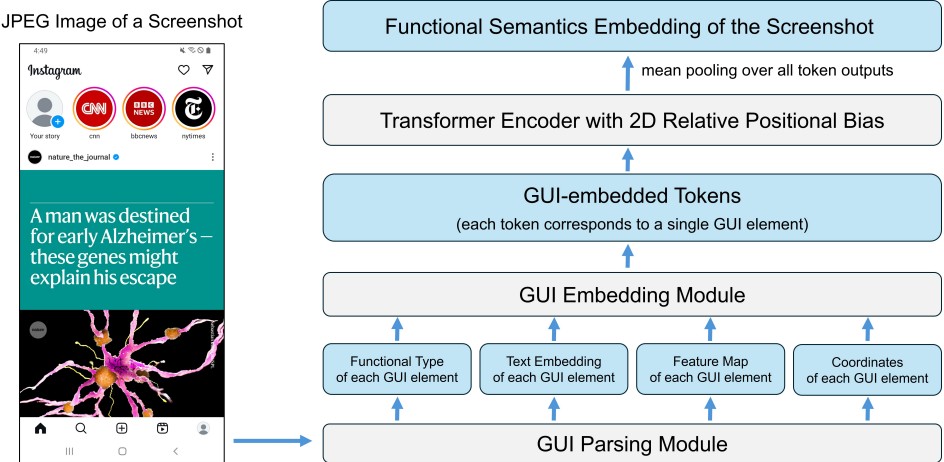

Figure 2: An overview of the Screen-SBERT framework. The GUI Parsing Module extracts modalities of GUI elements from a screenshot. The GUI Embedding Module then generates token representations. Finally, the tokens are processed by a Transformer encoder, and their outputs are aggregated via mean pooling to yield a final embedding that represents the functional semantics of the screenshot.

counterparts. This class imbalance reflects realistic conditions encountered when exploring mobile apps in industrial settings, thereby enhancing the applicability of our research.

A total of 1,814 screenshots across all apps were used for model training and evaluation. Detailed statistics and data collection procedures are provided in Appendix B.

## 5 SCREEN-SBERT

Our Screen-SBERT framework embeds the functional semantics of a screenshot without metadata such as the view hierarchy. Screen-SBERT consists of three key components: (1) the GUI Parsing Module, which extracts multimodal features from each GUI element; (2) the GUI Embedding Module, which encodes each GUI element into a unified token representation; (3) the Transformer Encoder, which contextualizes the embedded tokens; Figure 2 illustrates how these components interact to generate the final screenshot embedding.

To ensure the reproducibility and practical applicability of our framework, we deliberately avoided additional training in our default setting, such as pretraining the Transformer or fine-tuning the pretrained models used in the GUI Parsing Module. The only training we conducted was the contrastive learning procedure described in Section 6.

### 5.1 GUI PARSING MODULE

The GUI Parsing Module leverages publicly available pre-trained models to extract coordinates, vision feature maps, text embeddings, and functional types of GUI elements from a screenshot image. These models remain frozen during training. The overall processing pipeline of the GUI Parsing Module is illustrated in Figure 7 in Appendix D.

**GUI Detection:** First, an object detection model detects GUI elements and extracts their coordinates from the screenshot. In our experiments, we used the fine-tuned YOLOv8 model released by OmniParser (Lu et al., 2024). All detected elements were retained without filtering.

**Feature Map Extraction:** Previous works (Feiz et al., 2022; Wu et al., 2023) extracted intermediate feature maps from Faster R-CNN during GUI element detection. Similarly, our framework extracts the output feature maps of the final convolutional layer of the object detection model and crops the regions corresponding to each GUI element based on their coordinates. These cropped feature maps are then used as the vision modality for each GUI element.

**Text Embedding:** The output coordinates of each GUI element from the object detection model are also used to crop its image from the original screenshot. For each cropped GUI image, an OCR model—PaddleOCR-v2 (Du et al., 2021) in our case—recognizes text, and a Sentence-BERT model encodes the recognized text into an embedding. If the GUI image contains no text, its text embedding is assigned a zero vector.

**Functional Type Classification:** Previous works (Feiz et al., 2022; Fu et al., 2024) used pre-trained models to classify the type of each GUI element and treated the type as one of the modalities. However, we were unable to find any publicly available pre-trained classifier suitable for our study. Therefore, our framework uses the following heuristic approach as an alternative.

First, a fine-tuned image captioning model—Florence-2 (Xiao et al., 2024) released by OmniParser in our case—generates captions for GUI elements. For each GUI element, if the caption contains any predefined type name (e.g., "home," "back"), it is classified as that type; otherwise, it is classified as "other." The pseudocode and representative examples of this process are provided in Appendix F.

## 5.2 GUI EMBEDDING MODULE

The GUI Embedding Module converts each modality output from the GUI Parsing Module into embedding vectors of a shared dimensionality. These embeddings are summed element-wise to form a unified token representation for each GUI element. The overall processing pipeline of the GUI Embedding Module is illustrated in Figure 8 in Appendix E.

Each GUI element's functional type is transformed into an embedding vector $E_{\text{type}}$ by looking up a learnable embedding table. The text embedding and feature map are projected into the same dimensional space as $E_{\text{text}}$ and $E_{\text{vision}}$, respectively, through separate linear projection layers.

In our view, the layout structure—captured by the coordinates of GUI elements—is a key factor in distinguishing functional pages. We hypothesize that a fully connected linear projection layer is insufficient to effectively capture this coordinate information and that using embedding tables may be more effective. Accordingly, we followed the approach proposed in LayoutLM (Xu et al., 2020) and conducted an ablation study, as described in Subsection 8.3.

Specifically, the coordinates $(x_1, y_1, x_2, y_2)$—where $(x_1, y_1)$ denotes the top-left corner and $(x_2, y_2)$ the bottom-right corner—are expanded with the width $w = x_2 - x_1$ and height $h = y_2 - y_1$. Next, $x_1$, $x_2$, and $w$ are quantized to integers in $\{0, 1, \dots, 128\}$, and $y_1$, $y_2$, and $h$ are quantized to integers in $\{0, 1, \dots, 256\}$. These six quantized components are then embedded using separate learnable tables and summed with other modality embeddings as follows:

$$E_{\text{token}} = E_{\text{type}} + E_{\text{text}} + E_{\text{vision}} + E_{x_1} + E_{y_1} + E_{x_2} + E_{y_2} + E_w + E_h \tag{3}$$

## 5.3 TRANSFORMER ENCODER WITH 2D RELATIVE POSITIONAL BIAS

The token representations of GUI elements are processed by a Transformer encoder, and the outputs of all tokens are aggregated via mean pooling to form a final embedding representing the functional semantics of the screenshot.

Most previous works employed either relative positional encoding (RPE) or absolute positional encoding (APE), but not both. We hypothesize, however, that RPE and APE serve distinct roles in encoding GUI screenshots, and that combining them may lead to complementary effects: APE captures global layout structure, while RPE encodes pairwise spatial relationships between elements.

Therefore, in addition to the APE used in the GUI Embedding Module, we incorporate a 2D relative positional bias mechanism—inspired by the bucketing strategy of Raffel et al. (2020)—into a standard Transformer Encoder (Vaswani et al., 2017). Details of the bucketing mechanism and bias computation are provided in Appendix G.

## 6 CONTRASTIVE LEARNING

One of the key contributions of Sentence-BERT is the use of contrastive learning in a bi-encoder architecture to model sentence-level semantic similarity. Specifically, it optimizes a triplet loss:

$$\mathcal{L}_{\text{triplet}} = \max\left(\|f(a) - f(p)\| - \|f(a) - f(n)\| + \epsilon,\ 0\right) \tag{4}$$

Here, $a$ is the anchor, $p$ a positive sample, and $n$ a negative sample. $f(\cdot)$ denotes an embedding model. The loss pulls $a$ closer to $p$ and pushes it away from $n$ in the embedding space.

However, the performance of triplet learning heavily depends on mining effective $(a, p, n)$ combinations (Schroff et al., 2015). Due to time constraints, we did not develop an effective triplet mining strategy for our dataset. Instead, we adopted a supervised variant of the InfoNCE loss (van den Oord et al., 2019), in which a single positive sample from the same class and multiple negatives from different classes are randomly sampled using class annotations. This loss enables contrasting a single positive sample against multiple negatives for each anchor. The loss function is defined as:

$$\mathcal{L}_{\text{InfoNCE}} = -\log \frac{\exp\left(\text{sim}(f(s_a^{(i)}), f(s_p^{(i)}))/\tau\right)}{\exp\left(\text{sim}(f(s_a^{(i)}), f(s_p^{(i)}))/\tau\right) + \sum_{j \neq i} \exp\left(\text{sim}(f(s_a^{(i)}), f(s_n^{(j)}))/\tau\right)} \quad (5)$$

where $s_a^{(i)}$ is the anchor, $s_p^{(i)}$ is a positive sample drawn from the same class, and $s_n^{(j)}$ is a negative sample drawn from a different class. $\text{sim}(\cdot)$ denotes cosine similarity, and $\tau = 0.07$ is the temperature parameter.

When the anchor belongs to a singleton class such that a positive sample is unavailable, we applied a negative-only variant:

$$\mathcal{L}_{\text{negative-only}} = -\frac{1}{N-1} \sum_{j \neq i} \log\left(1 - \text{sim}(f(s_a^{(i)}), f(s_n^{(j)}))\right) \quad (6)$$

In our training strategy, each batch comprises one anchor, one positive sample, and multiple negative samples. Such a batch $\mathcal{B}$ can be formally represented as follows:

$$\mathcal{B} = \{s_a^{(i)}, s_p^{(i)}\} \cup \{s_n^{(j)} \mid 1 \leq j \leq N, j \neq i\} \quad (7)$$

# 7 EVALUATION

## 7.1 EVALUATION METHODOLOGY

To restate the problem addressed in this study, the task is to retrieve a functionally equivalent screenshot within the same app for a given query screenshot. We evaluated our framework by embedding all screenshots of the test apps and performing retrieval using cosine similarity–based nearest neighbor search. For each query screenshot, if the most similar neighbor belonged to the same class, it was counted as correct; otherwise incorrect. Based on this, we computed precision, recall, and F1 scores for each class and reported the macro average across classes. We additionally measured top-$k$ accuracy, considering a query correct if any of the $k$ nearest neighbors belonged to the same class.

We conducted all experiments under an out-of-domain (OOD) evaluation setting. Excluding ChatGPT-4o and Gemini 2.5 Pro, which do not require training, our framework and all baselines were trained on four apps—two social media and two shopping apps—and evaluated on two unseen apps—one social media and one shopping app, which were never used during training. While the test apps belong to similar application categories, their layout designs differ substantially from those in the training set, making this a valid OOD evaluation scenario.

## 7.2 BASELINES

We also implemented and trained several baseline models for comparison with our framework. All baselines and our framework were trained and evaluated under identical app configurations and training/validation splits. (However, ChatGPT-4o and Gemini 2.5 Pro were accessed through commercial APIs and therefore required no training; they were evaluated under the same configurations.) All experiments, including both baselines and our framework, were conducted on a single NVIDIA GeForce RTX 3090. The list of baselines is as follows:

**CLIP ViT-Base:** We first fine-tuned a vision–language model (VLM) using the contrastive learning strategy described in Section 6. CLIP-Base represents the most practical choice for contrastive learning in an environment restricted to a single GPU. Accordingly, our evaluation of CLIP-Base aims to answer the question: "How effectively can contrastive learning with a smaller VLM perform under constrained GPU memory resources?"

**Screen Correspondence (Wu et al., 2023) and PW2SS (Fu et al., 2024):** These two frameworks extract the modalities of GUI elements from a screenshot, treating the entire screenshot as a sentence and each modality as a word token. They are trained exclusively with MLM, without any supervised signals. Their evaluation addresses the following question: "In the absence of large-scale public datasets, how well can an MLM-only training perform in few-shot learning on small datasets?"

**Screen Similarity Transformer (Feiz et al., 2022):** This model serves as our primary baseline, as its task definition and screenshot similarity criteria are closely aligned with ours. It adopts a cross-encoder architecture in which two screenshots are jointly fed into the model, and a similarity logit is derived from the [CLS] token. The logit is optimized using a binary cross-entropy (BCE) loss. Its evaluation addresses the question: "What are the performance differences between cross-encoder and bi-encoder architectures in capturing screenshot similarity?"

**ChatGPT-4o and Gemini 2.5 Pro:** These multimodal large language models (MLLMs) were evaluated using commercial APIs without any additional training. For each test screenshot, we provided an MLLM with a fixed natural language prompt along with the screenshot and requested two outputs: (1) **Detail**—a one-sentence description of the functionalities available in the screenshot; and (2) **Concision**—a concise page class name obtained by compressing the description into a short phrase. We then encoded these natural language outputs using an mGTE (Zhang et al., 2024). The subsequent evaluation procedure was identical to that of our framework. This evaluation addresses the following question: "To what extent can a natural language–based retrieval approach, powered by well-trained commercial MLLMs, demonstrate its potential?"

CLIP, Screen Correspondence, and PW2SS were evaluated in the same manner as our framework—by embedding all screenshots of the test apps and conducting retrieval via cosine similarity–based nearest neighbor search. In contrast, the Screen Similarity Transformer cannot produce consistent embeddings for individual screenshots, as it employs a cross-encoder architecture. Therefore, we evaluated it by feeding every (query, candidate) pair into the model to obtain similarity logits, followed by nearest neighbor search based on these logits.

## 7.3 RESULTS

| | Macro | | | Top-$k$ Accuracy | | |
|---|---|---|---|---|---|---|
| | Precision | Recall | F1 | $k=1$ | $k=2$ | $k=3$ |
| CLIP ViT-B/32 | 0.792 | 0.779 | 0.772 | 0.844 | 0.889 | 0.896 |
| CLIP ViT-B/16 | 0.688 | 0.697 | 0.678 | 0.767 | 0.818 | 0.860 |
| Screen Correspondence | 0.634 | 0.612 | 0.614 | 0.683 | 0.723 | 0.753 |
| PW2SS | 0.721 | 0.709 | 0.707 | 0.769 | 0.801 | 0.809 |
| Screen Similarity Transformer | 0.856 | 0.827 | 0.832 | 0.854 | 0.886 | 0.905 |
| ChatGPT-4o (Concision) | 0.495 | 0.478 | 0.454 | 0.518 | 0.591 | 0.615 |
| ChatGPT-4o (Detail) | 0.704 | 0.642 | 0.658 | 0.685 | 0.732 | 0.768 |
| Gemini 2.5 Pro (Concision) | 0.529 | 0.504 | 0.482 | 0.571 | 0.662 | 0.700 |
| Gemini 2.5 Pro (Detail) | 0.806 | 0.797 | 0.796 | 0.797 | 0.826 | 0.857 |
| **Screen-SBERT (Ours)** | **0.920** | **0.892** | **0.901** | **0.921** | **0.938** | **0.942** |

Table 1: Comparison of screen retrieval performance between the baselines and our framework.

Table 1 reports the test results on X and Temu after training each model on Instagram, Facebook, Amazon Shopping, and Coupang. Results for additional app configurations are provided in Tables 7–14 in Appendix J. A comprehensive analysis of these tables indicates that our Screen-SBERT consistently outperforms the baselines, except for the Screen Similarity Transformer, in terms of Macro F1 and Top-1 Accuracy. Although the Screen Similarity Transformer surpasses our framework in certain configurations, our bi-encoder design offers a clear advantage in computational efficiency compared to its cross-encoder architecture, as discussed in Section 2.

Regarding the results of the MLLM-based approaches, we observed a notable improvement when encoding detailed descriptions rather than concise outputs. This is likely because compressing a screenshot description into a short phrase introduces substantial information loss.

However, even with the detailed descriptions, the overall performance remained comparable to that of the other baselines and fell short of our framework. One plausible explanation is that the natural-language outputs of the MLLMs had to be re-encoded by an mGTE model, during which additional information may have been lost. If embedding vectors could be obtained directly from these models without passing through natural language generation, the retrieval performance might have been higher. Nonetheless, such a setup would be conceptually similar to scaling up our framework and the other baselines with larger training data.

## 8 ABLATION STUDIES

Our framework integrates multiple components, each of which may affect the overall performance. To isolate the contribution of individual components, we conducted three types of ablation studies. All ablation studies followed the same setting: the model was trained on Instagram, Facebook, Amazon Shopping, and Coupang, and evaluated on X and Temu.

### 8.1 COMPARISON OF TRAINING OBJECTIVES

| | Macro | | | Top-$k$ Accuracy | | |
|---|---|---|---|---|---|---|
| | Precision | Recall | F1 | $k = 1$ | $k = 2$ | $k = 3$ |
| BCE | 0.681 | 0.686 | 0.674 | 0.743 | 0.784 | 0.797 |
| MLM Only | 0.892 | 0.845 | 0.859 | 0.859 | 0.890 | 0.894 |
| MLM + CL | 0.900 | 0.869 | 0.880 | 0.898 | 0.932 | 0.934 |
| **CL Only (Default)** | **0.920** | **0.892** | **0.901** | **0.921** | **0.938** | **0.942** |

Table 2: Comparison of training objectives for screen retrieval in our framework.

First, we trained our framework under three alternative settings: (1) binary cross-entropy (BCE) loss for equivalence classification; (2) masked language modeling (MLM) without any supervised signals; and (3) MLM pretraining followed by fine-tuning with contrastive learning (CL).

As summarized in Table 2, our default setting—training exclusively with contrastive learning—achieved the best performance. Notably, performance declined when MLM pretraining was applied, suggesting that MLM-based objectives are less effective with limited training data, since they require large-scale datasets to learn meaningful contextual representations.

### 8.2 ABLATION ON MODALITIES

| | Macro | | | Top-$k$ Accuracy | | |
|---|---|---|---|---|---|---|
| | Precision | Recall | F1 | $k = 1$ | $k = 2$ | $k = 3$ |
| w/o functional type | 0.855 | 0.844 | 0.846 | 0.887 | 0.911 | 0.916 |
| w/o text embedding | 0.915 | 0.873 | 0.885 | 0.887 | 0.919 | 0.928 |
| w/o vision feature map | 0.860 | 0.836 | 0.840 | 0.864 | 0.917 | 0.924 |
| w/o coordinates | 0.779 | 0.762 | 0.762 | 0.813 | 0.841 | 0.868 |
| **Default Setting** | **0.920** | **0.892** | **0.901** | **0.921** | **0.938** | **0.942** |

Table 3: Effect of removing each modality on screen retrieval performance.

We next evaluated the contribution of each modality for token representation by selectively removing one modality at a time. As shown in Table 3, all modalities contributed positively to performance, with the coordinates proving to be the most critical. This highlights the importance of spatial layout information in modeling the functional semantics of screenshots.

## 8.3 COMPARISON OF POSITIONAL ENCODING APPROACHES

|  |  | Macro | | | Top-$k$ Accuracy | | |
|---|---|---|---|---|---|---|---|
|  |  | Precision | Recall | F1 | $k = 1$ | $k = 2$ | $k = 3$ |
| w/o APE | w/o RPE | 0.736 | 0.727 | 0.727 | 0.800 | 0.816 | 0.846 |
|  | with RPE | 0.779 | 0.762 | 0.762 | 0.813 | 0.841 | 0.868 |
| Linear Projection | w/o RPE | 0.787 | 0.769 | 0.772 | 0.821 | 0.829 | 0.858 |
|  | with RPE | 0.840 | 0.834 | 0.834 | 0.848 | 0.881 | 0.902 |
| Embedding Tables | w/o RPE | 0.870 | 0.848 | 0.855 | 0.905 | 0.934 | 0.940 |
|  | with RPE | **0.920** | **0.892** | **0.901** | **0.921** | **0.938** | **0.942** |

Table 4: Comparison of positional encoding strategies for screen retrieval.

To validate the hypotheses introduced in Subsections 5.2 and 5.3, we conducted experiments comparing different positional encoding strategies. Specifically, we tested three configurations of absolute positional encoding (APE): (1) removing the coordinates modality; (2) projecting the coordinates vector through a fully connected layer; and (3) embedding each coordinate component with a separate learnable lookup table. Each configuration was evaluated with and without RPE in the Transformer Encoder, resulting in six experimental variants in total.

Table 4 summarizes the results. Among the three APE configurations, the embedding-table approach yielded the highest performance, significantly outperforming the fully connected linear projection method. This suggests that simple linear projection is insufficient for capturing the layout structure of GUI elements, and that APE is effective only in the form of learnable embedding tables.

Furthermore, applying RPE improved performance across all APE configurations. As we expected, RPE complemented APE, yielding additional performance gains.

## 9 QUALITATIVE RESULTS

To qualitatively assess whether each model constructs an embedding space that effectively captures the functional semantics of screenshots, we computed pairwise similarities between all screenshots to construct a similarity matrix and visualized it using the t-SNE algorithm. The figure on the right illustrates the results of our Screen-SBERT and the Screen Similarity Transformer for the Temu app. Further qualitative results for other models and apps can be found in Appendix K.

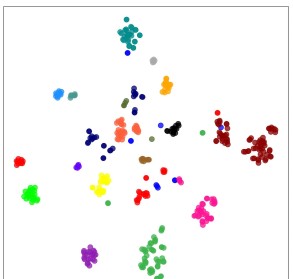 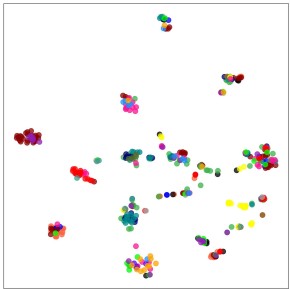

Screen-SBERT (Ours)    Similarity Transformer

From this visualization, we observed that the cross-encoder structure of the Screen Similarity Transformer does not produce clear clusters that separate functional pages. This suggests that while pairwise binary classification with BCE loss can determine whether two screenshots are equivalent, it fails to construct a meaningful embedding space for GUI screenshots that captures functional semantics beyond pairwise decisions. In contrast, contrastive learning explicitly optimizes for relative similarity, enabling the model to build a more structured, semantically meaningful embedding space.

In conclusion, by synthesizing the findings from Sections 7, 8, and 9, we summarize the key insights of this study as follows: (1) Contrastive learning with a bi-encoder architecture is more practical than pairwise classification with a cross-encoder for GUI screenshot retrieval; (2) The effectiveness of CL is maintained even under limited training data, unlike MLM; (3) Capturing the overall layout structure using GUI element coordinates is essential for embedding functional semantics; (4) When using absolute embedding of GUI element coordinates, lookup table–based APE is substantially

more effective than fully connected linear projection; and (5) Combining RPE and APE leads to complementary effects that improve performance.

## 10 LIMITATIONS AND FUTURE WORKS

Building on the key insights summarized above, we conclude this paper by outlining some limitations and potential directions for future work.

**Subjectivity in Page Class Annotation:** In our dataset, the annotation of page classes was performed solely based on our intuitive judgment, which may introduce a degree of subjectivity into the boundaries between classes.

To partially assess the consistency and degree of subjectivity in these annotations, we developed a simple verification tool and recruited nine external participants to perform the validation task. Please refer to Appendix C for details of this assessment of subjectivity.

Independently of this evaluation, experimental results in out-of-domain settings suggest that our method does not merely memorize superficial layout patterns but instead implicitly learns the underlying grouping criteria reflected in our annotations. This implies that the framework may be capable of flexibly adapting to varying annotation schemes across users or contexts.

Of course, this hypothesis requires further validation. As future work, we plan to construct alternative annotation sets independently generated by multiple annotators and assess whether our method can maintain consistent retrieval performance across these diverse subjective groupings.

**Few-Shot Learning on Small-Scale Dataset Only:** Because no large-scale public dataset exists for the problem addressed in this study, our experiments fall under few-shot learning. It remains unclear whether the performance observed in this setting would generalize to large-scale scenarios, and MLM could potentially yield better performance when trained on larger datasets.

Nevertheless, this study is significant in demonstrating the feasibility of effectively performing few-shot learning under data-constrained settings. Unlike prior works that assume access to large-scale datasets, our approach highlights a practical learning strategy that remains applicable when resources are severely limited.

**Absence of Larger VLM Experiments:** Due to the limitation of training on a single GPU, we were only able to include CLIP-Base as a baseline and could not perform experiments involving contrastive fine-tuning of larger VLMs with multiple GPUs. While this restriction may have capped the upper bound of achievable performance, it also suggests that our framework can provide a practical option in environments where access to extensive computational resources is not feasible. Future research should investigate whether scaling to larger VLMs can yield further performance improvements and examine how such models compare with our approach, potentially revealing scaling laws for embedding the functional semantics of GUI screenshots.

**Practical Application in GUI Agents:** A particularly promising direction for future work is applying our proposed method to real-world GUI agents. Once the functional semantics of a GUI screenshot are understood and its page class is identified, incorporating prior knowledge of that page into the prompt can improve the agent's task success rate. Simple one-step decision simulations of this idea are provided in Appendix L. We believe that refining the perspective illustrated in the appendix into a more concrete and sophisticated system could further enhance the performance of GUI agents.

Considering these limitations and future directions, our study opens up broad research opportunities, including architectural extensions of the framework and its integration with downstream tasks. As an early exploratory study, we hope this work contributes to advancing the use of functional semantics of screenshots in future GUI agent systems.

## ETHICS STATEMENT

Due to the nature of social media and shopping applications, the original screenshots used in this study may contain privacy-sensitive information. Therefore, releasing the raw screenshots from

these domains is not feasible, which is also one of the main reasons for the lack of large-scale public datasets suitable for the problem we address.

Instead, we release the modality data obtained by preprocessing all screenshots through our GUI Parsing Module. In these data, the vision feature maps and text embeddings—encoded with YOLO and Sentence-BERT, respectively—are not invertible to reconstruct the original screenshots or expose privacy-sensitive information. This ensures that the released data can be shared safely without privacy concerns.

The example figures in this paper contain only screenshots that were carefully selected to ensure that no privacy-sensitive information is exposed.

## REPRODUCIBILITY STATEMENT

All code and datasets used to produce the experimental results in this paper are available at `https://github.com/conference-anonymous-author/ICLR2026_Screen_SBERT`. The repository contains the dataset labeled with page classes, the PyTorch implementation of our framework, the training and evaluation code, and the weight files of our pre-trained models. The provided dataset and training code allow users to readily reproduce our experiments and retrain the model.

Note that the released dataset does not include the original screenshots; instead, it provides the preprocessed outputs of each screenshot obtained through the GUI Parsing Module.

As the dataset was primarily collected through manual effort, the collection code is not included in the repository. Details of the dataset collection process can be found in Appendix B.

## THE USE OF LARGE LANGUAGE MODELS

First, we used ChatGPT-4o to survey the related works cited in this paper. Its Deep Research functionality was initially employed to compile a list of recent studies, after which we directly examined and analyzed those works to determine our research topic and direction.

Second, we drafted the manuscript in our native language and then translated it into English with the assistance of ChatGPT-4o. The LLM's role was strictly limited to translation and did not extend to determining the research direction, formulating hypotheses, or designing experiments.

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

# A  MORE EXAMPLES OF FUNCTIONALLY EQUIVALENCE

In the following figures, screenshots enclosed in the same rectangle are functionally equivalent and thus belong to the same class, whereas those in different rectangles are functionally different and belong to distinct classes.

## A.1  INSTAGRAM

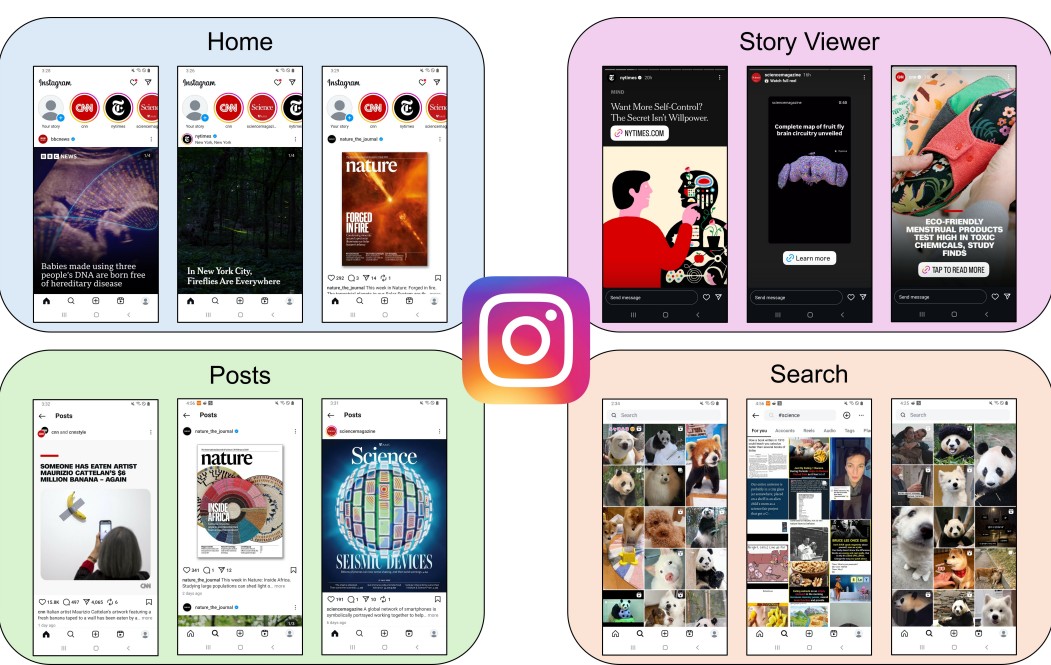

## A.2  FACEBOOK

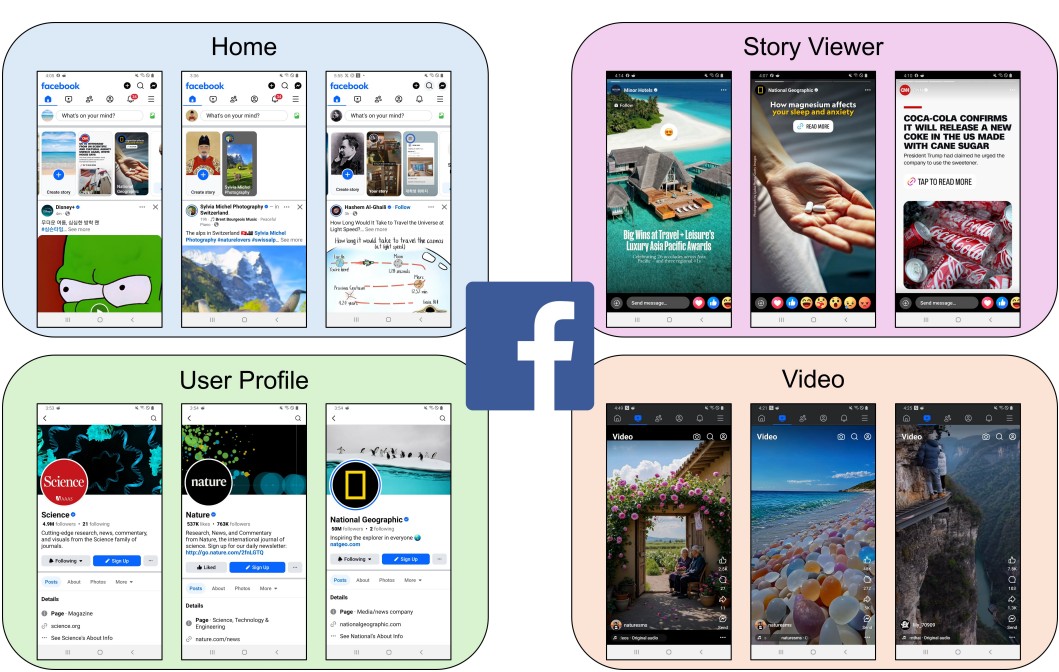

## A.3   X

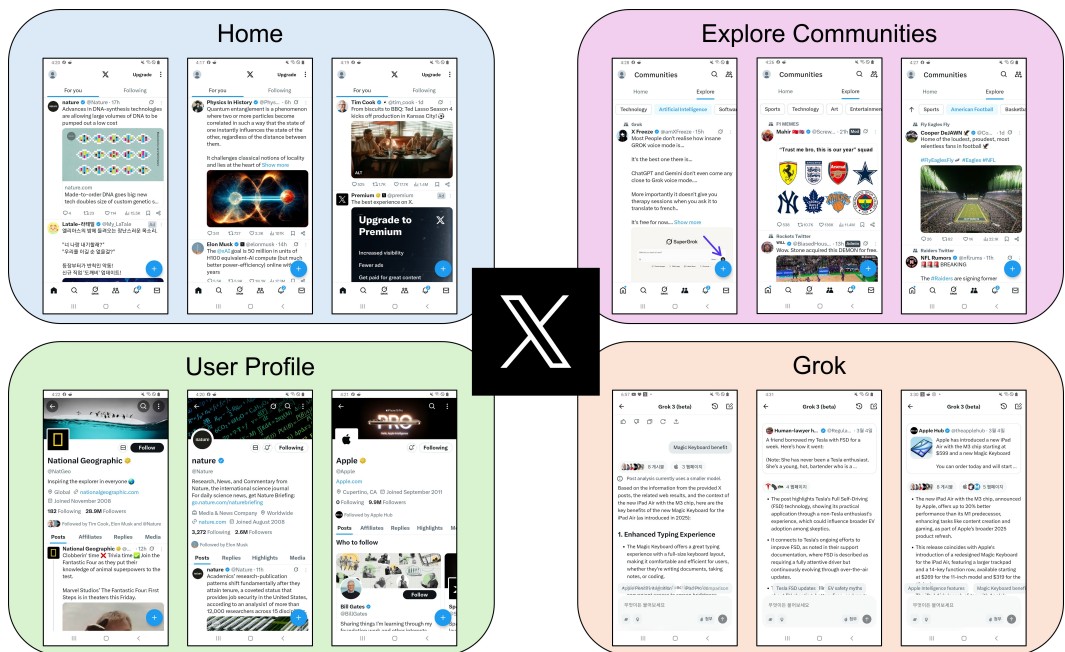

## A.4   AMAZON SHOPPING

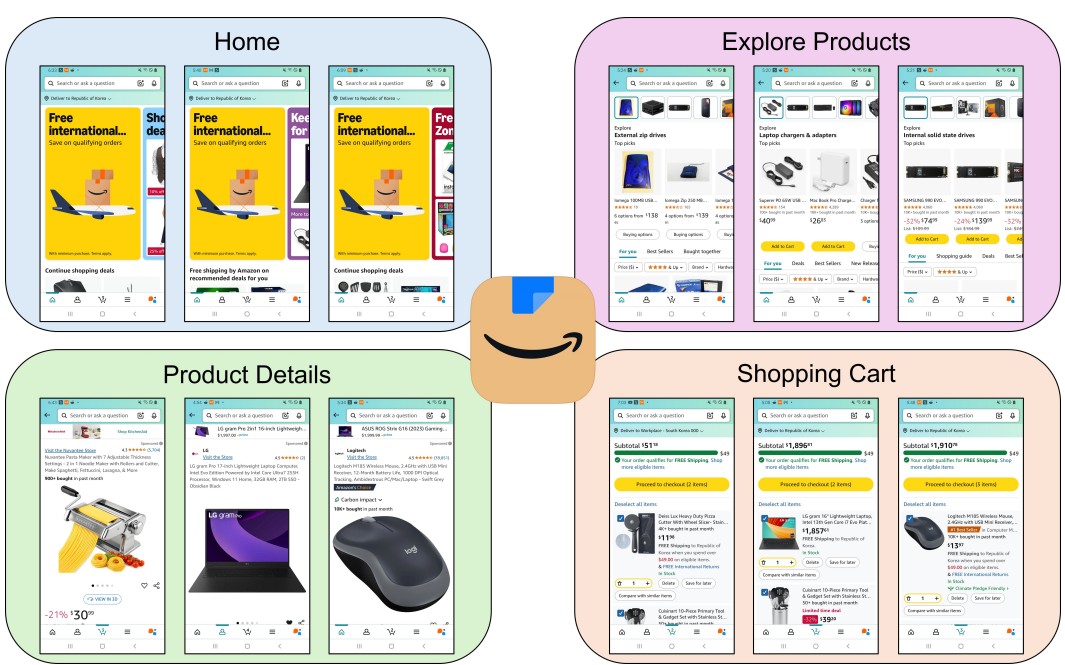

## A.5   COUPANG

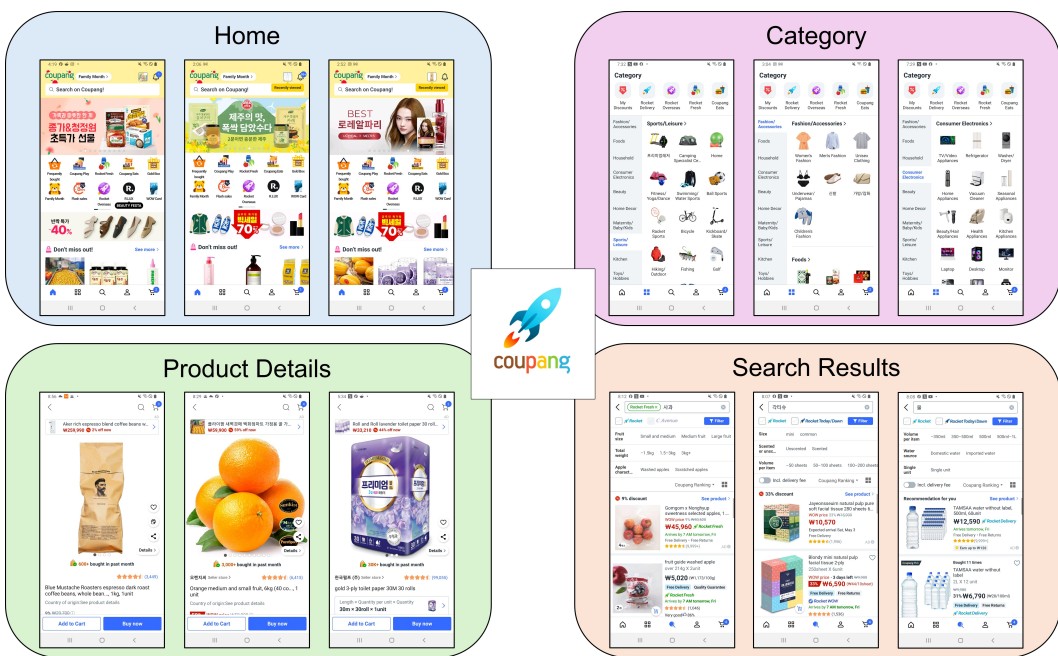

## A.6   TEMU

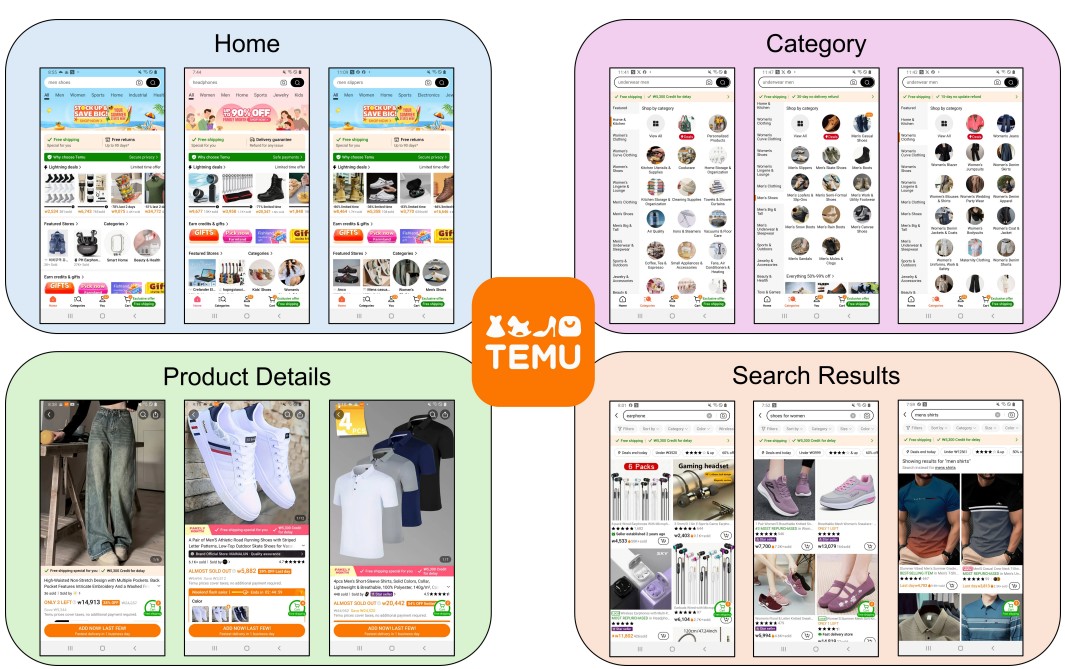

# B DETAILS OF DATASET

## B.1 DATA COLLECTION

To construct our dataset, we aimed to include as many page classes as possible. We treated the home screen as the root node and manually constructed a screen transition tree. For each screen, we applied a pretrained object detection model—YOLOv8 in our case—to identify candidate GUI elements. We then manually tapped each detected element, and if the element turned out to be clickable, we captured the resulting screen and added it as a child node in the transition tree. In some cases, the resulting screen was functionally equivalent to its parent. For example, tapping a "like" or "follow" button may update the appearance of the button without causing a transition to a different page class. Nevertheless, such screens were still included in the transition tree. This exploration was conducted in a breadth-first manner, capturing all reachable screens at a given depth before proceeding to the next level.

We constructed a single screen transition tree for each app without repeated exploration. While the exploration was not exhaustive, a breadth-first tree with depth 3 was sufficient to capture most user-reachable pages in typical usage flows.

All captured screens were used for model training and evaluation without any additional filtering based on layout or semantics, thereby preserving all observed screens, including those with visual or functional redundancy. This reflects the conditions under which GUI agents are deployed in real-world industrial settings.

We manually annotated all collected screenshots with class labels. Specifically, screenshots that are functionally equivalent were grouped together and assigned a label summarizing their functionality. The total number of classes (i.e., groups) was not predetermined; during the manual inspection process, if a screenshot could not be categorized into any existing group, a new group was created.

## B.2 STATISTICS

|  | Total Number of Screenshots | Number of Page Classes |
| --- | --- | --- |
| Instagram | 167 | 58 |
| Facebook | 301 | 100 |
| X | 331 | 86 |
| Amazon Shopping | 127 | 25 |
| Coupang | 393 | 73 |
| Temu | 495 | 64 |

Table 5: Number of screenshots and page classes for each app.

As a result of this data collection procedure, we constructed the dataset summarized in Table 5. Instagram and Amazon Shopping featured relatively simpler layout structures compared to the other apps, with fewer GUI elements per screen. Consequently, their screen transition trees were smaller, and significantly fewer screenshots were collected. In contrast, apps such as Coupang and Temu exhibited more complex layouts with a larger number of GUI elements, resulting in more screenshots being included in the dataset.

Figure 3 illustrates the distribution of screenshots across page classes for each app. As shown in the figure, our dataset has a clear class imbalance. This kind of imbalance is common in real-world industrial settings, indicating that our experimental setup closely reflects practical deployment scenarios.

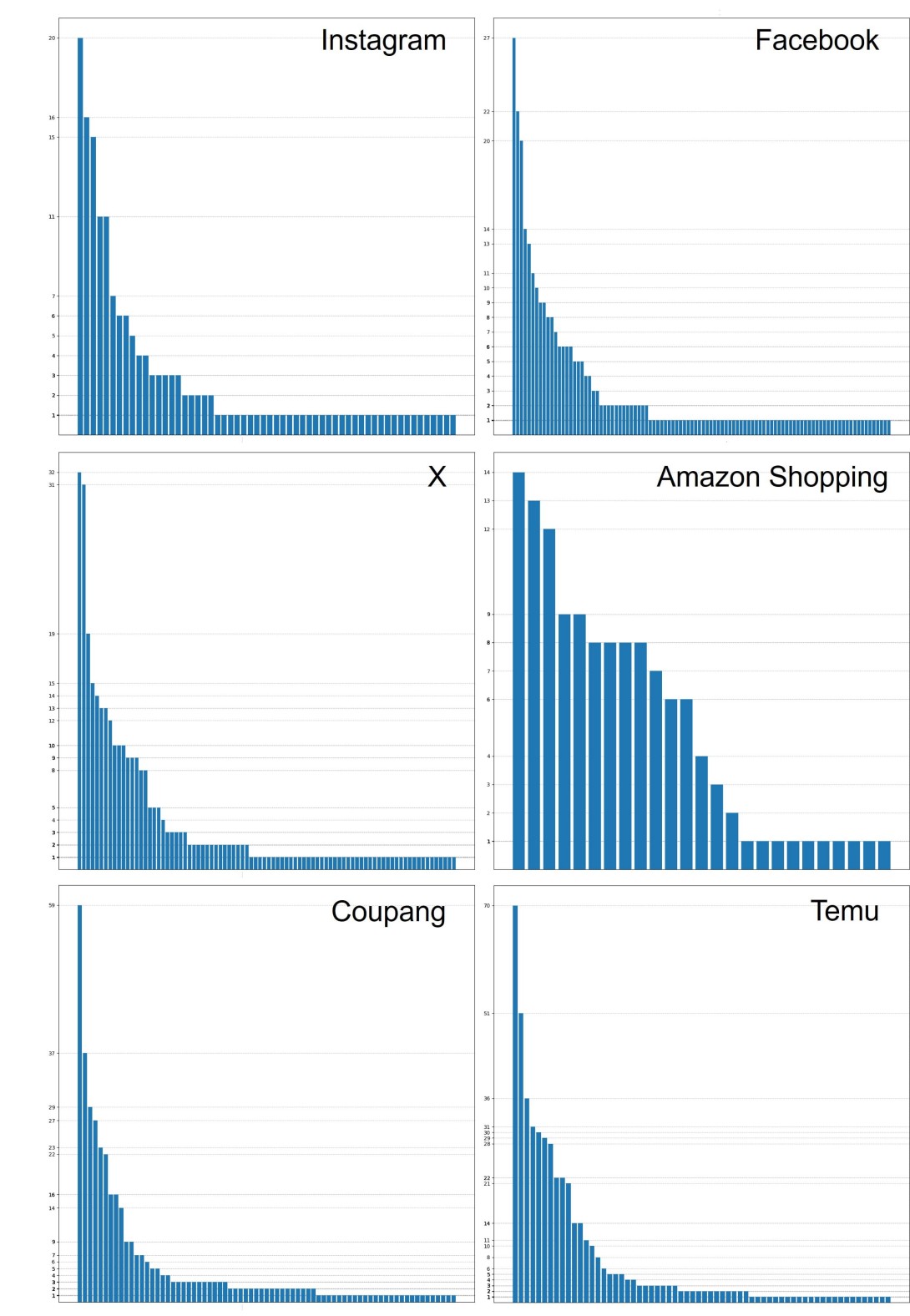

Figure 3: Distribution of screenshots across page classes for each app. Each bar corresponds to a page class, and the height of the bar indicates the number of screenshots assigned to that class. The x-axis represents the index of page classes, sorted in descending order by frequency.

# C ASSESSMENT OF SUBJECTIVITY IN PAGE CLASS ANNOTATION

When designing our framework, we placed strong emphasis on encoding the global layout and the positional relationships among GUI elements. Our intuition is that humans typically rely on such layout structure and contextual relationships when identifying the functional semantics of a GUI screenshot. In particular, the paradigm adopted in this work—as well as in several baselines—treats the entire screenshot as a "sentence" and each GUI element as a "word." This perspective reflects our view that humans primarily perceive a GUI screenshot by interpreting the global layout and relational organization of its elements, rather than by analyzing individual components in isolation.

This intuition served as the primary criterion for our page-class annotation. To evaluate the extent to which others agree with our assigned page classes, we recruited nine external validators who were not involved in any part of model design or experimentation and participated solely in this annotation-verification task. Their role was to independently assess whether our assigned page classes were appropriate, allowing us to examine how subjective our page-class annotation actually is and to quantify the degree of inter-rater agreement achieved in practice.

## C.1 PARTICIPANTS

Given the sensitive personal information contained in our dataset, the participants involved in the validation process were limited to our personal acquaintances with whom sharing such information would not pose a significant concern. The majority of participants were non-experts in machine learning and were not familiar with its theoretical foundations. For this reason, we expect that they approached the task without any ML-related preconceptions and evaluated the functional equivalence of screenshots from the perspective of ordinary app users.

## C.2 DETAILS OF VERIFICATION

After receiving a sufficient explanation of the concepts of functional equivalence and page class as defined in our study—as well as the criteria by which screenshots had been grouped based on these definitions—the validators proceeded with the review task. They examined every screenshot in the dataset individually and indicated, using a binary value of 0 (disagree) or 1 (agree), whether they considered the assigned page class to be appropriate.

We asked them to record their agreement according to the following guidelines: (i) If a screenshot is judged not to be functionally equivalent to the other screenshots grouped under the same page class, the validator should mark 0 (disagree) for each such screenshot; (ii) If an entire page class appears to require full integration into another class—i.e., if two page classes that are functionally equivalent seem to have been unnecessarily separated—the validator should mark 0 (disagree) for all screenshots belonging to that class.

Finally, for each app, we calculated the agreement rate as the proportion of screenshots marked as 1 (agree) among all screenshots.

## C.3 SURVEY

In addition, we conducted several surveys to supplement the agreement-rate metric. Participants were asked to respond to the following questions using a 5-point Likert scale.

**General Questions**

1) I have previously studied machine learning theory.

2) I have prior experience performing dataset-validation or annotation tasks similar to this one.

3) I clearly understood the concepts of functional equivalence and page class, the structure of the dataset, and the criteria used to group screenshots.

**App-specific Questions**

1) I have used this app before.

2) I can easily infer the available functionalities from each screenshot in this app.

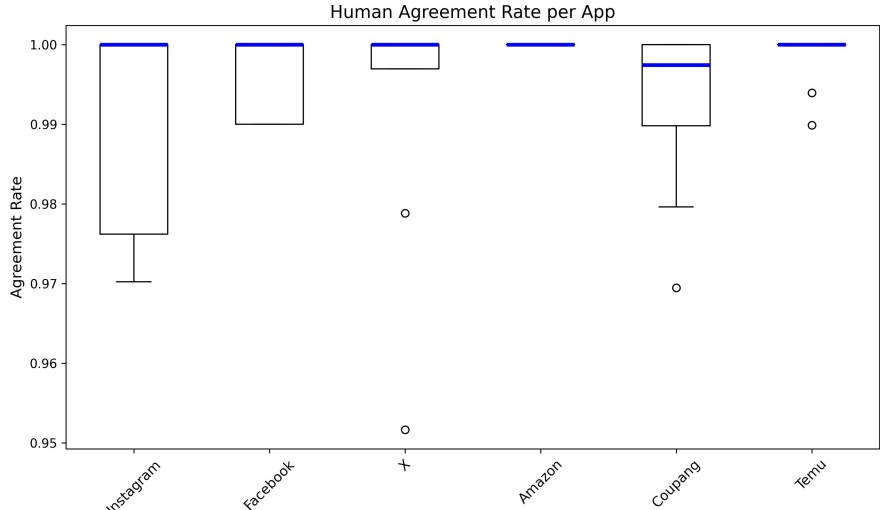

Figure 4: Box plots of participants' agreement rates across the six apps.

## C.4 RESULTS

### C.4.1 PARTICIPANTS' AGREEMENT RATE

Figure 4 presents the distribution of agreement rates for each app using box plots. For each box, the blue horizontal line indicates the median agreement, while the top and bottom edges correspond to the third quartile and first quartile, respectively. The height of the box therefore represents the interquartile range (IQR), which captures the middle 50% of the participants' agreement scores. The whiskers extend to the largest and smallest values within $1.5 \times$ IQR of the quartile boundaries, and any observations outside this range are marked as outliers (shown as individual points).

This box plot visualization shows that participants largely agreed with our page class annotations, and even in cases where they disagreed with certain screenshots, such disagreements did not exceed 5% of the data for any app. This indicates that our subjective judgment of the boundaries between page classes is broadly aligned with typical app users' perspectives.

### C.4.2 RESPONSES TO THE GENERAL SURVEY QUESTIONS

Figure 5 visualizes the responses to the general survey questions using pie charts. These results indicate that the majority of participants were non-experts with no background in machine learning theory, suggesting that their agreement judgments were made from the perspective of typical app users, without theoretical preconceptions (e.g., assumptions about whether the annotations would be effective for model training). In addition, participants reported that they clearly understood our definitions of functional equivalence and page class, suggesting that these concepts are intuitive even to non-experts.

### C.4.3 RESPONSES TO THE APP-SPECIFIC SURVEY QUESTIONS

Figure 6 visualizes the responses to the app-specific survey questions using box plots. When considered together with Figure 4, an interesting tendency emerges: participants who were less familiar with a given app—those with little or no prior experience using it—tended to exhibit higher agreement with the provided annotations. Conversely, participants who regularly used an app or were more familiar with it showed a slightly wider dispersion in their agreement scores.

We interpret this tendency as arising from differences in the information participants relied upon when evaluating the validity of the existing annotations. Although we introduced the concepts of functional equivalence and page classes during the instruction phase and provided several illustrative examples, it was not feasible to comprehensively cover all boundary cases, as each app contains

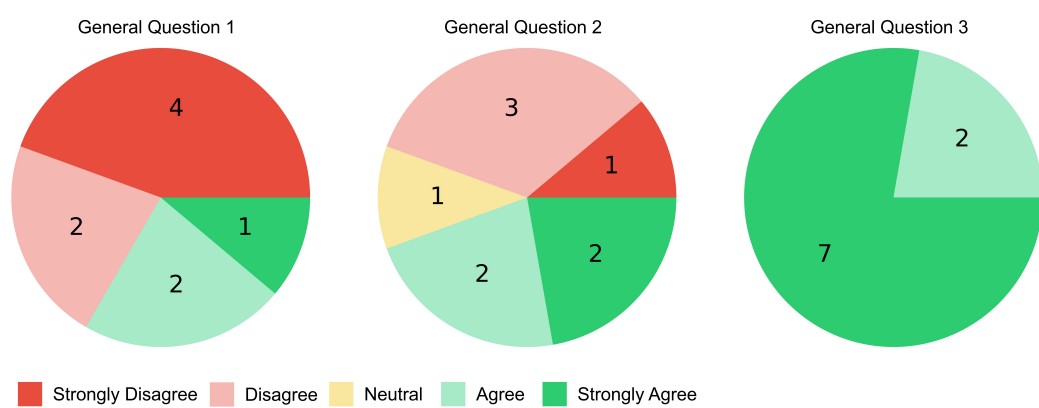

Q1) I have previously studied machine learning theory.
Q2) I have prior experience performing dataset-validation or annotation tasks similar to this one.
Q3) I clearly understood the concepts of functional equivalence and page class,
    the structure of the dataset, and the criteria used to group screenshots.

Figure 5: Pie charts summarizing the responses to the three general survey questions.

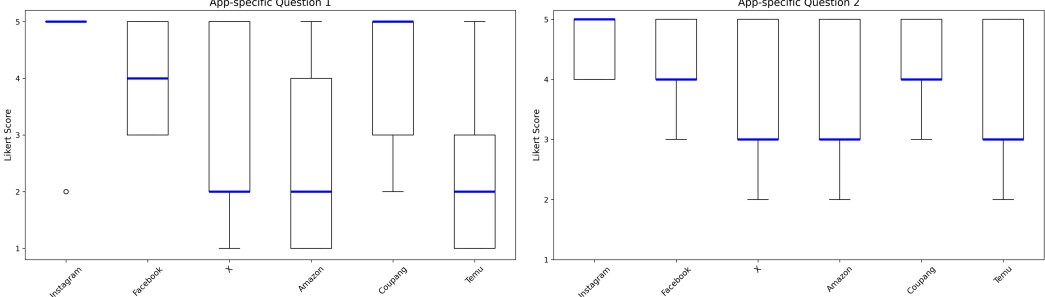

Q1) I have used this app before.
Q2) I can easily infer the available functionalities from each screenshot in this app.

Figure 6: Box plots showing the distribution of responses to the two app-specific survey questions.

dozens of page classes. In this setting, participants who were highly familiar with a particular app often incorporated their own usage experiences—beyond the formal definitions—into their decisions. Because such experience-driven interpretations naturally vary from person to person, they can lead to modestly lower or more variable agreement for apps with which participants are familiar.

In contrast, participants with little or no prior experience had fewer personal priors to draw upon and therefore relied more consistently on the definitions and examples provided during the instruction phase. With fewer idiosyncratic interpretations affecting their judgments, their decisions tended to be more stable and aligned.

Importantly, despite these experience-driven variations, the overall level of disagreement remained below 5% across the entire dataset. This indicates that while minor differences related to individual usage histories do occur in a small subset of cases, our functional page-class definition is generally stable and applicable to the vast majority of screens.

In future work, this subjectivity could be further reduced by expanding the annotation criteria with more fine-grained guidelines or a richer set of illustrative examples. Such additions would help refine the definition and provide clearer decision-making cues for annotators, thereby strengthening the robustness of future datasets.

## D    OVERVIEW OF THE GUI PARSING MODULE

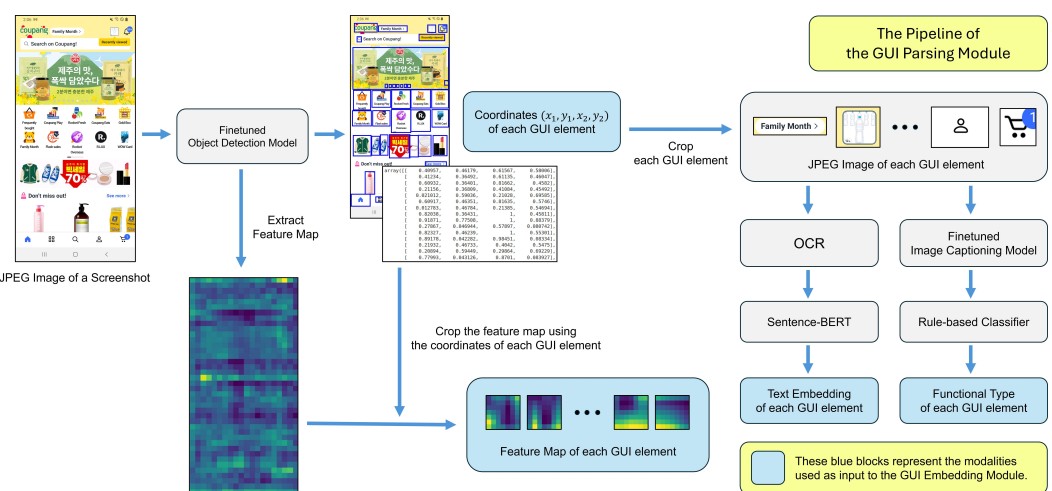

Figure 7: An overview of the GUI Parsing Module. A finetuned object detection model detects GUI elements and extracts their coordinates and feature maps. For each GUI element, cropped image regions are used to obtain text embeddings via OCR and Sentence-BERT, and functional type indices via image captioning and rule-based classification. The output includes coordinates, feature maps, text embeddings, and functional types of all GUI elements.

## E    OVERVIEW OF THE GUI EMBEDDING MODULE

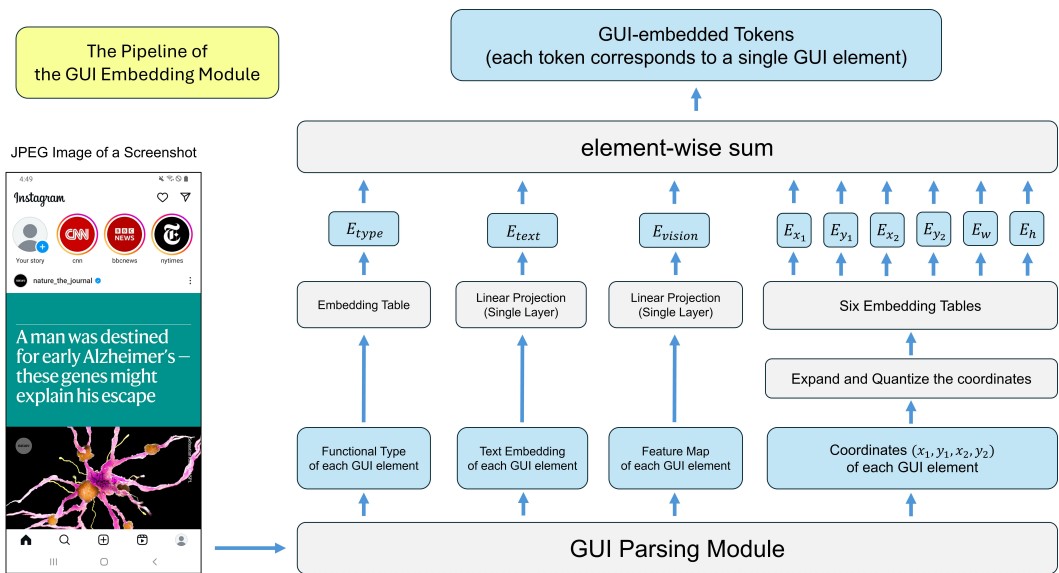

Figure 8: An overview of the GUI Embedding Module. For each GUI element, modality embeddings—corresponding to functional type, text embedding, vision feature map, and coordinates—are transformed into same-dimensional vectors and combined via element-wise summation to produce a unified token representation.

## F   RULE-BASED CLASSIFICATION OF FUNCTIONAL TYPES

### F.1   PSEUDOCODE

---
**Algorithm 1** Rule-based Classification of Functional Types

---
1: **procedure** CLASSIFYFUNCTIONALTYPE($cropped\_GUI\_image$, $functional\_types$)
2:     $caption \leftarrow$ ImageCaptioningModel($cropped\_GUI\_image$)
3:     **for all** $type \in functional\_types$ **do**
4:         **if** $caption$ contains $type$ **then**
5:             **return** $type$
6:         **end if**
7:     **end for**
8:     **return** `"other"`
9: **end procedure**

---

### F.2   EXAMPLES

Figure 9 presents representative examples of each functional type. However, not all GUI elements in our dataset were classified as cleanly as these examples. The primary reason lies in the limited captioning performance of the Florence-2 model we used. When the image captioning model generates inaccurate descriptions, the heuristic classification procedure described in Algorithm 1 may lead to incorrect results.

This heuristic classification approach was adopted as a practical alternative due to the lack of a publicly available GUI type classifier and the limited resources available for manually labeling a training dataset. If a high-performing GUI type classifier were to be developed and integrated, it would likely offer a more robust solution and further improve the overall performance of the framework.

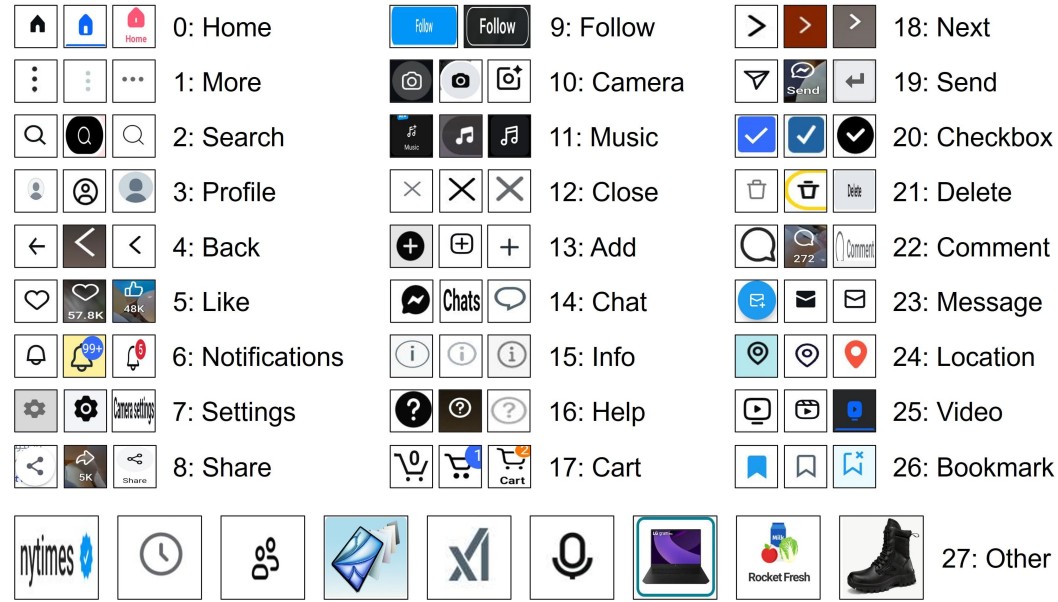

Figure 9: Representative examples of each functional type.

## G  DETAILS OF 2D RELATIVE POSITIONAL ENCODING

In our Transformer Encoder, the attention score between two GUI tokens is computed as follows:

$$\text{score}_{i,j} = \frac{Q_i \cdot K_j^\top}{\sqrt{d}} + \text{bias}_x(\text{bucket}(\Delta x)) + \text{bias}_y(\text{bucket}(\Delta y)) \tag{8}$$

Here, $\text{bias}_x$ and $\text{bias}_y$ are learnable embedding tables that map quantized horizontal and vertical distances, respectively, to scalar biases that are added to the attention logits in each head. $\Delta x = x_i - x_j$ and $\Delta y = y_i - y_j$ represent the differences between the center coordinates of the two GUI elements. The bucketing function $\text{bucket}(\Delta)$ is defined to quantize $\Delta$ using a log-scale scheme:

$$\text{bucket}(\Delta) = \text{sign}(\Delta) \cdot \min(\lfloor \log(\max(|\Delta|, \epsilon)) \rfloor, N-1) + (N-1) \tag{9}$$

where $\text{sign}(\cdot)$ is a function that returns the sign of inputs, $\epsilon$ is a small constant to prevent undefined behavior when $\Delta = 0$ in the logarithmic term, and $N$ determines the bucket range, with $2N-1$ rows in the bias tables. In our experiments, $N$ was fixed at 32.

These biases help the attention layer explicitly learn the relative positional relationships between GUI elements. For instance, they allow the model to capture that the Home button is positioned more to the left than the Cart button, or that the profile image is located higher than the Follow button. Without such biases, the model must rely on implicitly learning these relationships, with no guarantee of success. By encoding them explicitly, overall performance can be improved.

## H  DETAILS OF BASELINES (NON-MLLM MODELS)

Since the source codes for the previous works used as baselines—particularly the pretrained models for extracting GUI element modalities—were not available, we implemented each model ourselves based on the descriptions in their papers. We replaced these unavailable components with publicly available models released by OmniParser, which were also used in our own framework. In other words, the coordinates, vision feature maps, text embeddings, and types of GUI elements in each model were extracted in the same way as in our framework. Aside from these unavoidable modifications, we adhered to the original descriptions as closely as possible. We tuned and experimented with multiple variants of each model by varying their sizes and training hyperparameters, and reported in the experimental results the version that achieved the best performance on our dataset.

### H.1  SCREEN CORRESPONDENCE

This model represents each GUI element with up to three tokens—its type, vision feature map, and text embedding—which are processed by a Transformer encoder. The output tokens corresponding to the same element are then aggregated through mean pooling to produce a single embedding per element. Finally, the overall screenshot embedding is obtained by mean pooling across all element embeddings.

This model does not use absolute positional encoding (APE) but instead adopts relative positional encoding. The model is trained exclusively with a masked language modeling (MLM) objective, without any additional supervised objectives.

### H.2  PW2SS

This model takes as input to the Transformer the text embedding of the entire screenshot and the type embeddings of individual GUI elements. In addition, it incorporates the Layout Encoder from Liu et al. (2018), a multi-layer CNN that encodes layout images, whose output embedding is included as an additional token in the input sequence. Finally, lookup table–based APE is added to the text and GUI type tokens.

The model is trained solely with MLM. After training, the final embedding of the entire screenshot is computed by applying max pooling over all token outputs.

### H.3 SCREEN SIMILARITY TRANSFORMER

This model adopts a cross-encoder architecture, where both screenshots are fed simultaneously into a single Transformer. For each GUI element extracted from the screenshots, a token is formed by combining its vision feature map, linearly projected coordinates, type embedding, and a screen embedding that denotes which screenshot the element belongs to. These tokens, together with a [CLS] token used to compute the similarity logit, constitute the input sequence to the model.

The model is trained to perform same/different classification for screenshot pairs by optimizing the similarity logit from the [CLS] token with a binary cross-entropy (BCE) loss.

### H.4 MODEL CONFIGURATIONS

The tables below present the hyperparameter configurations for each model. Optimal values for training hyperparameters, such as learning rate and batch size, may vary depending on the model architecture and dataset. We therefore tested multiple configurations and report the results for the version that achieved the best performance.

The number of epochs may vary depending on the loss function and dataset. For each configuration, we trained the model for a sufficient number of epochs to ensure convergence on the training data and selected the checkpoint that achieved the best performance on the validation set for evaluation. The validation set was created by partitioning 15% of the page classes from the training apps; while the app composition was identical to that of the training data, the page classes were disjoint.

|  | CLIP ViT-B/32 | CLIP ViT-B/16 | Screen Correspondence |
|---|---|---|---|
| Attention Heads |  |  | 4 |
| Attention Layers | Same as the official | Same as the official | 4 |
| Hidden Size | release version | release version | 256 |
| Dropout Rate |  |  | 0.3 |
| Optimizer | AdamW | AdamW | AdamW |
| Learning Rate | 1e-5 | 1e-5 | 1e-4 |
| Weight Decay | 1e-2 | 1e-2 | 1e-2 |
| Loss Function | InfoNCE | InfoNCE | Mean Squared Error |
| Total Epochs | 5 | 5 | 200 |
| Batch Size | Different for each app | Different for each app | 8 |

|  | PW2SS | Similarity Transformer | Screen-SBERT |
|---|---|---|---|
| Attention Heads | 4 | 8 | 8 |
| Attention Layers | 4 | 6 | 6 |
| Hidden Size | 256 | 256 | 768 |
| Dropout Rate | 0.3 | 0.3 | 0.3 |
| Optimizer | AdamW | AdamW | AdamW |
| Learning Rate | 1e-4 | 2e-4 | 2e-5 |
| Weight Decay | 1e-2 | 1e-2 | 1e-2 |
| Loss Function | Mean Squared Error | Binary Cross Entropy | InfoNCE |
| Total Epochs | 200 | 100 | 5 |
| Batch Size | 8 | 8 | Different for each app |

Table 6: Hyperparameter configurations of each model.

# I  INPUT PROMPT AND OUTPUT EXAMPLES FOR MLLM-BASED BASELINES

## I.1  INPUT PROMPT

In our experiments, ChatGPT-4o and Gemini 2.5 Pro received the following Chain-of-Thought–style prompt (Wei et al., 2022) together with a single screenshot image. This prompt was refined through multiple rounds of trial and error to achieve the strongest retrieval performance from the MLLMs.

---

You are an AI assistant specialized in identifying the functional page of a mobile GUI screenshot.

### Background ###
Functional Equivalence: Two screens are considered functionally equivalent if a user can access the same functionalities through them, regardless of differences in displayed content. Conversely, two screens are functionally different if they provide different functionalities, even when they look visually similar.
Functional Page Class: We define a 'functional page' as a class and a 'screenshot' as an instance. A page class is a group of screenshots that are functionally equivalent—meaning that all screenshots in the class allow the user to perform the same set of functions.
Formally, a page class c consists of multiple screenshots, written as:
c = { s_1, s_2, ..., s_k }
where c denotes the page class and each s_i represents the i-th screenshot belonging to that class.

### Examples ###
In social media apps, examples of functional page classes include 'Home', 'Story Viewer', 'Posts', and 'User Profile', among many others. In shopping apps, examples include 'Home', 'Category', 'Search Results', and 'Product Details', along with many more variants.

### Instruction ###
Based on the descriptions above, your task is to generate a functional page class name for the given screenshot. Your decision must be based solely on the functionalities available on the screen, ignoring the specific content displayed. The examples above are illustrative only; do not reuse them unless they truly match the functionality of the given screen.

### Output Format ###
Follow the format strictly and include every section header exactly as written.

### Observation ###
Provide a detailed account of the functional GUI elements visible on the current screen. Exclude the content of products or posts; focus only on GUI components that enable specific user actions.

### Thought ###
Describe your reasoning that leads to the final answer.

### Detailed Description ###
Summarize in exactly one sentence what overall functions the current screen allows the user to perform.

### Page Class ###
Provide a concise name for the functional page class.

---

ChatGPT-4o and Gemini 2.5 Pro generated outputs following the output format specified in the prompt above. The text produced in the **### Detailed Description ###** section and the **### Page Class ###** section was then encoded using an mGTE model and used for retrieval evaluation.

## I.2 OUTPUT EXAMPLES

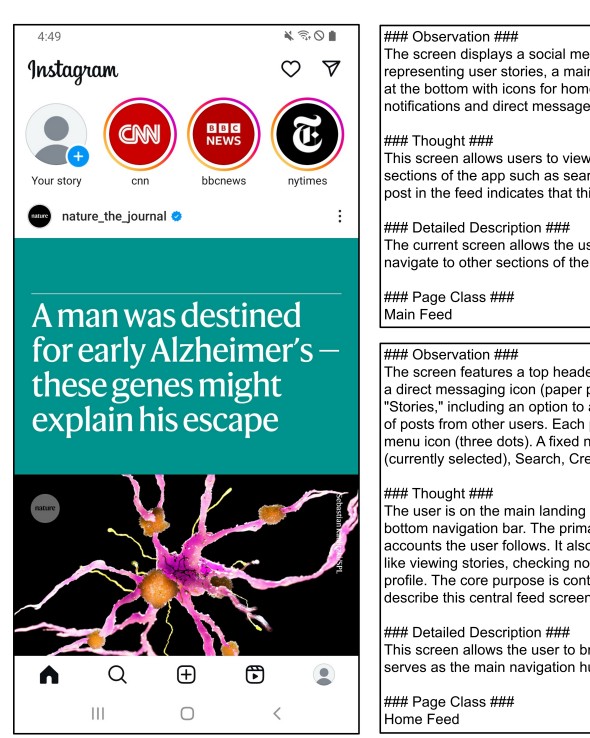

### Observation ###
The screen displays a social media interface with several key elements: a row of circular icons at the top representing user stories, a main feed area showing a post with an image and text, and a navigation bar at the bottom with icons for home, search, create post, reels, and profile. There are also icons for notifications and direct messages at the top.

### Thought ###
This screen allows users to view and interact with the main feed, access stories, and navigate to other sections of the app such as search, post creation, reels, and their profile. The presence of stories and a post in the feed indicates that this is the primary interface for consuming content.

### Detailed Description ###
The current screen allows the user to view the main feed, interact with posts, access stories, and navigate to other sections of the app.

### Page Class ###
Main Feed

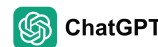 ChatGPT

### Observation ###
The screen features a top header with the application logo ("Instagram"), a notifications icon (heart), and a direct messaging icon (paper plane). Below the header is a horizontally scrollable bar for viewing "Stories," including an option to add to "Your story." The main content area is a vertically scrollable feed of posts from other users. Each post has a header with the poster's profile picture and username, and a menu icon (three dots). A fixed navigation bar is present at the bottom of the screen with icons for Home (currently selected), Search, Create New Post, Reels, and User Profile.

### Thought ###
The user is on the main landing screen of the application, as indicated by the selected "Home" icon in the bottom navigation bar. The primary function of this screen is to display a scrollable feed of posts from accounts the user follows. It also serves as a central hub, providing access to other key functionalities like viewing stories, checking notifications, sending messages, creating content, and accessing their profile. The core purpose is content consumption in a feed format. Therefore, the page class should describe this central feed screen.

### Detailed Description ###
This screen allows the user to browse a primary feed of posts and stories from followed accounts and serves as the main navigation hub for accessing other core features of the application.

### Page Class ###
Home Feed

Gemini

Figure 10: MLLM Outputs for Instagram's Home Page

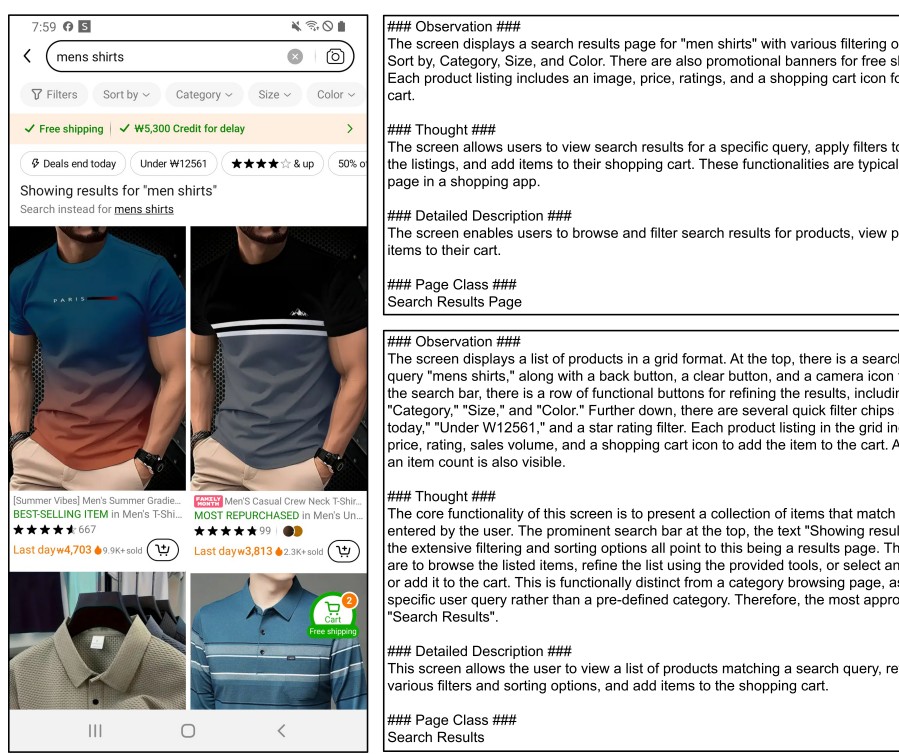

### Observation ###
The screen displays a search results page for "men shirts" with various filtering options such as Filters, Sort by, Category, Size, and Color. There are also promotional banners for free shipping and credits. Each product listing includes an image, price, ratings, and a shopping cart icon for adding items to the cart.

### Thought ###
The screen allows users to view search results for a specific query, apply filters to refine the results, sort the listings, and add items to their shopping cart. These functionalities are typical of a search results page in a shopping app.

### Detailed Description ###
The screen enables users to browse and filter search results for products, view product details, and add items to their cart.

### Page Class ###
Search Results Page

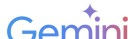 ChatGPT

### Observation ###
The screen displays a list of products in a grid format. At the top, there is a search bar pre-filled with the query "mens shirts," along with a back button, a clear button, and a camera icon for visual search. Below the search bar, there is a row of functional buttons for refining the results, including "Filters," "Sort by," "Category," "Size," and "Color." Further down, there are several quick filter chips such as "Deals end today," "Under W12561," and a star rating filter. Each product listing in the grid includes an image, title, price, rating, sales volume, and a shopping cart icon to add the item to the cart. A floating cart icon with an item count is also visible.

### Thought ###
The core functionality of this screen is to present a collection of items that match a specific search query entered by the user. The prominent search bar at the top, the text "Showing results for 'men shirts'", and the extensive filtering and sorting options all point to this being a results page. The user's primary actions are to browse the listed items, refine the list using the provided tools, or select an item to view its details or add it to the cart. This is functionally distinct from a category browsing page, as it is generated by a specific user query rather than a pre-defined category. Therefore, the most appropriate classification is "Search Results".

### Detailed Description ###
This screen allows the user to view a list of products matching a search query, refine these results using various filters and sorting options, and add items to the shopping cart.

### Page Class ###
Search Results

Gemini

Figure 11: MLLM Outputs for Temu's Search Results Page

# J OUT-OF-DOMAIN EXPERIMENTS FOR ALL COMBINATIONS OF TRAINING/TEST APP SETS

Tables 7 –14 present the results of experiments with varying training/test app configurations. In each setup, one social media app and one shopping app are designated as test apps, while the remaining four apps are used for training. Since the two test apps are entirely unseen during training, each configuration constitutes a valid out-of-domain evaluation. All other experimental settings are identical to those described in Section 7.

Figure 12 summarizes the results across all nine OOD configurations. The vertical lines represent the performance range from the lowest to the highest score for each model, and the square marker denotes the median performance.

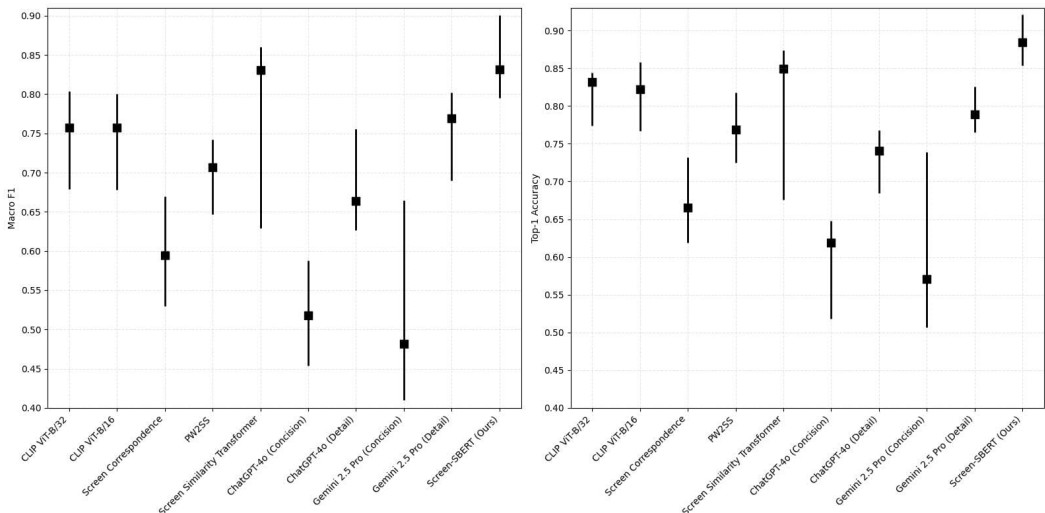

Figure 12: Left: Macro F1. Right: Top-1 Accuracy.

|  | Macro | | | Top-$k$ Accuracy | | |
|---|---|---|---|---|---|---|
|  | Precision | Recall | F1 | $k = 1$ | $k = 2$ | $k = 3$ |
| CLIP ViT-B/32 | 0.729 | 0.745 | 0.728 | 0.830 | 0.844 | 0.875 |
| CLIP ViT-B/16 | 0.686 | 0.691 | 0.685 | 0.850 | 0.870 | 0.900 |
| Screen Correspondence | 0.557 | 0.539 | 0.542 | 0.663 | 0.725 | 0.735 |
| PW2SS | 0.706 | 0.686 | 0.688 | 0.818 | 0.847 | 0.858 |
| Screen Similarity Transformer | **0.894** | 0.818 | 0.839 | 0.849 | 0.904 | **0.922** |
| ChatGPT-4o (Concision) | 0.585 | 0.579 | 0.574 | 0.646 | 0.673 | 0.735 |
| ChatGPT-4o (Detail) | 0.760 | 0.759 | 0.756 | 0.759 | 0.797 | 0.833 |
| Gemini 2.5 Pro (Concision) | 0.683 | 0.663 | 0.665 | 0.739 | 0.811 | 0.821 |
| Gemini 2.5 Pro (Detail) | 0.805 | 0.776 | 0.778 | 0.806 | 0.865 | 0.871 |
| **Screen-SBERT (Ours)** | 0.857 | **0.861** | **0.856** | **0.894** | **0.914** | 0.918 |

Table 7: Evaluation on **Instagram** and **Amazon Shopping**.

|  | Macro | | | Top-$k$ Accuracy | | |
|---|---|---|---|---|---|---|
|  | Precision | Recall | F1 | $k=1$ | $k=2$ | $k=3$ |
| CLIP ViT-B/32 | 0.737 | 0.716 | 0.720 | 0.827 | 0.851 | 0.858 |
| CLIP ViT-B/16 | 0.780 | 0.769 | 0.763 | 0.858 | 0.874 | 0.880 |
| Screen Correspondence | 0.611 | 0.586 | 0.585 | 0.665 | 0.716 | 0.769 |
| PW2SS | 0.732 | 0.698 | 0.698 | 0.725 | 0.755 | 0.788 |
| Screen Similarity Transformer | 0.767 | 0.700 | 0.714 | 0.741 | 0.810 | 0.879 |
| ChatGPT-4o (Concision) | 0.624 | 0.581 | 0.588 | 0.648 | 0.687 | 0.732 |
| ChatGPT-4o (Detail) | 0.763 | 0.726 | 0.731 | 0.768 | 0.821 | 0.844 |
| Gemini 2.5 Pro (Concision) | 0.544 | 0.498 | 0.485 | 0.565 | 0.698 | 0.740 |
| Gemini 2.5 Pro (Detail) | 0.769 | 0.776 | 0.767 | 0.791 | 0.821 | 0.836 |
| **Screen-SBERT (Ours)** | **0.801** | **0.800** | **0.797** | **0.877** | **0.898** | **0.907** |

Table 8: Evaluation on **Instagram** and **Coupang**.

|  | Macro | | | Top-$k$ Accuracy | | |
|---|---|---|---|---|---|---|
|  | Precision | Recall | F1 | $k=1$ | $k=2$ | $k=3$ |
| CLIP ViT-B/32 | 0.810 | 0.807 | 0.804 | 0.840 | 0.857 | 0.876 |
| CLIP ViT-B/16 | 0.794 | 0.814 | 0.800 | 0.831 | 0.857 | 0.899 |
| Screen Correspondence | 0.622 | 0.606 | 0.607 | 0.619 | 0.690 | 0.715 |
| PW2SS | 0.761 | 0.725 | 0.734 | 0.755 | 0.807 | 0.826 |
| Screen Similarity Transformer | 0.883 | 0.857 | 0.860 | 0.874 | 0.931 | 0.934 |
| ChatGPT-4o (Concision) | 0.503 | 0.499 | 0.478 | 0.530 | 0.590 | 0.647 |
| ChatGPT-4o (Detail) | 0.688 | 0.645 | 0.650 | 0.685 | 0.740 | 0.771 |
| Gemini 2.5 Pro (Concision) | 0.601 | 0.578 | 0.567 | 0.626 | 0.683 | 0.697 |
| Gemini 2.5 Pro (Detail) | 0.775 | 0.771 | 0.769 | 0.771 | 0.808 | 0.827 |
| **Screen-SBERT (Ours)** | **0.903** | **0.887** | **0.889** | **0.905** | **0.937** | **0.939** |

Table 9: Evaluation on **Instagram** and **Temu**.

|  | Macro | | | Top-$k$ Accuracy | | |
|---|---|---|---|---|---|---|
|  | Precision | Recall | F1 | $k=1$ | $k=2$ | $k=3$ |
| CLIP ViT-B/32 | 0.770 | 0.737 | 0.745 | 0.832 | 0.886 | 0.883 |
| CLIP ViT-B/16 | 0.787 | 0.782 | 0.779 | 0.842 | **0.893** | 0.895 |
| Screen Correspondence | 0.531 | 0.538 | 0.530 | 0.648 | 0.721 | 0.762 |
| PW2SS | 0.768 | 0.742 | 0.741 | 0.799 | 0.831 | 0.867 |
| **Screen Similarity Transformer** | **0.866** | **0.846** | **0.850** | **0.868** | 0.890 | **0.898** |
| ChatGPT-4o (Concision) | 0.486 | 0.464 | 0.458 | 0.619 | 0.686 | 0.776 |
| ChatGPT-4o (Detail) | 0.645 | 0.628 | 0.631 | 0.741 | 0.779 | 0.838 |
| Gemini 2.5 Pro (Concision) | 0.447 | 0.434 | 0.428 | 0.545 | 0.668 | 0.712 |
| Gemini 2.5 Pro (Detail) | 0.713 | 0.689 | 0.690 | 0.777 | 0.844 | 0.855 |
| Screen-SBERT (Ours) | 0.824 | 0.813 | 0.813 | 0.855 | 0.865 | 0.868 |

Table 10: Evaluation on **Facebook** and **Amazon Shopping**.

| | Macro | | | Top-$k$ Accuracy | | |
|---|---|---|---|---|---|---|
| | Precision | Recall | F1 | $k = 1$ | $k = 2$ | $k = 3$ |
| CLIP ViT-B/32 | 0.789 | 0.770 | 0.772 | 0.836 | 0.850 | 0.870 |
| CLIP ViT-B/16 | 0.748 | 0.732 | 0.733 | 0.821 | 0.861 | 0.875 |
| Screen Correspondence | 0.625 | 0.589 | 0.596 | 0.690 | 0.736 | 0.752 |
| PW2SS | 0.748 | 0.718 | 0.721 | 0.792 | 0.826 | 0.836 |
| **Screen Similarity Transformer** | **0.846** | **0.832** | **0.831** | **0.861** | **0.879** | 0.880 |
| ChatGPT-4o (Concision) | 0.537 | 0.532 | 0.518 | 0.642 | 0.701 | 0.764 |
| ChatGPT-4o (Detail) | 0.688 | 0.660 | 0.664 | 0.739 | 0.781 | 0.813 |
| Gemini 2.5 Pro (Concision) | 0.452 | 0.421 | 0.410 | 0.530 | 0.666 | 0.722 |
| Gemini 2.5 Pro (Detail) | 0.756 | 0.750 | 0.741 | 0.789 | 0.836 | 0.852 |
| Screen-SBERT (Ours) | 0.804 | 0.799 | 0.795 | 0.854 | 0.870 | **0.882** |

Table 11: Evaluation on **Facebook** and **Coupang**.

| | Macro | | | Top-$k$ Accuracy | | |
|---|---|---|---|---|---|---|
| | Precision | Recall | F1 | $k = 1$ | $k = 2$ | $k = 3$ |
| CLIP ViT-B/32 | 0.776 | 0.753 | 0.757 | 0.829 | 0.875 | 0.889 |
| CLIP ViT-B/16 | 0.779 | 0.758 | 0.764 | 0.822 | 0.884 | 0.904 |
| Screen Correspondence | 0.681 | 0.674 | 0.670 | 0.708 | 0.738 | 0.756 |
| PW2SS | 0.705 | 0.683 | 0.683 | 0.752 | 0.807 | 0.841 |
| Screen Similarity Transformer | 0.829 | 0.823 | 0.822 | 0.837 | 0.879 | 0.886 |
| ChatGPT-4o (Concision) | 0.489 | 0.462 | 0.455 | 0.548 | 0.624 | 0.676 |
| ChatGPT-4o (Detail) | 0.657 | 0.616 | 0.627 | 0.689 | 0.733 | 0.780 |
| Gemini 2.5 Pro (Concision) | 0.448 | 0.434 | 0.417 | 0.507 | 0.601 | 0.653 |
| Gemini 2.5 Pro (Detail) | 0.725 | 0.705 | 0.708 | 0.765 | 0.816 | 0.837 |
| **Screen-SBERT (Ours)** | **0.866** | **0.842** | **0.849** | **0.884** | **0.893** | **0.908** |

Table 12: Evaluation on **Facebook** and **Temu**.

| | Macro | | | Top-$k$ Accuracy | | |
|---|---|---|---|---|---|---|
| | Precision | Recall | F1 | $k = 1$ | $k = 2$ | $k = 3$ |
| CLIP ViT-B/32 | 0.687 | 0.718 | 0.694 | 0.806 | 0.817 | 0.849 |
| CLIP ViT-B/16 | 0.751 | 0.743 | 0.741 | 0.816 | 0.862 | 0.883 |
| Screen Correspondence | 0.570 | 0.567 | 0.562 | 0.732 | 0.782 | 0.802 |
| PW2SS | 0.650 | 0.652 | 0.647 | 0.747 | 0.796 | 0.848 |
| Screen Similarity Transformer | 0.709 | 0.616 | 0.629 | 0.676 | 0.718 | 0.760 |
| ChatGPT-4o (Concision) | 0.576 | 0.552 | 0.549 | 0.619 | 0.693 | 0.705 |
| ChatGPT-4o (Detail) | 0.699 | 0.697 | 0.692 | 0.765 | 0.794 | 0.825 |
| Gemini 2.5 Pro (Concision) | 0.551 | 0.522 | 0.519 | 0.637 | 0.723 | 0.740 |
| Gemini 2.5 Pro (Detail) | 0.810 | 0.807 | 0.802 | 0.826 | 0.856 | 0.900 |
| **Screen-SBERT (Ours)** | **0.848** | **0.830** | **0.832** | **0.870** | **0.888** | **0.901** |

Table 13: Evaluation on **X** and **Amazon Shopping**.

| | Macro | | | Top-$k$ Accuracy | | |
|---|---|---|---|---|---|---|
| | Precision | Recall | F1 | $k = 1$ | $k = 2$ | $k = 3$ |
| CLIP ViT-B/32 | 0.701 | 0.694 | 0.679 | 0.774 | 0.838 | 0.860 |
| CLIP ViT-B/16 | 0.759 | 0.765 | 0.757 | 0.815 | 0.854 | 0.861 |
| Screen Correspondence | 0.594 | 0.609 | 0.595 | 0.659 | 0.701 | 0.714 |
| PW2SS | 0.776 | 0.737 | 0.742 | 0.773 | 0.792 | 0.796 |
| Screen Similarity Transformer | **0.840** | 0.717 | 0.746 | 0.741 | 0.768 | 0.788 |
| ChatGPT-4o (Concision) | 0.582 | 0.556 | 0.548 | 0.602 | 0.648 | 0.686 |
| ChatGPT-4o (Detail) | 0.691 | 0.672 | 0.671 | 0.717 | 0.762 | 0.789 |
| Gemini 2.5 Pro (Concision) | 0.505 | 0.455 | 0.436 | 0.537 | 0.674 | 0.717 |
| Gemini 2.5 Pro (Detail) | 0.761 | 0.760 | 0.754 | 0.785 | 0.815 | 0.847 |
| **Screen-SBERT (Ours)** | 0.800 | **0.807** | **0.799** | **0.860** | **0.875** | **0.884** |

Table 14: Evaluation on **X** and **Coupang**.

## K ADDITIONAL QUALITATIVE RESULTS

Figures 13 - 18 present t-SNE visualizations of the pairwise similarities among screenshots of each app. In these qualitative results, the differences between the embedding models—CLIP, Screen Correspondence, PW2SS, and our Screen-SBERT—are not clearly distinguishable. By contrast, the results of the Screen Similarity Transformer highlight a notable limitation: despite its cross-encoder architecture achieving high Macro F1 and Top-1 Accuracy scores in retrieval tests, it fails to construct a meaningful embedding space for GUI screenshots that reflects functional semantics. This shortcoming may limit its applicability to downstream tasks beyond pairwise equivalence classification.

### K.1 INSTAGRAM AND AMAZON SHOPPING

Figures 13 and 14 show t-SNE visualizations of the similarity matrices for Instagram and Amazon Shopping produced by each model. All models were trained on the other four apps—Facebook, X, Coupang, and Temu. For clarity, only classes with at least three screenshots are displayed; the remaining classes were used in the t-SNE computation but omitted from the plots.

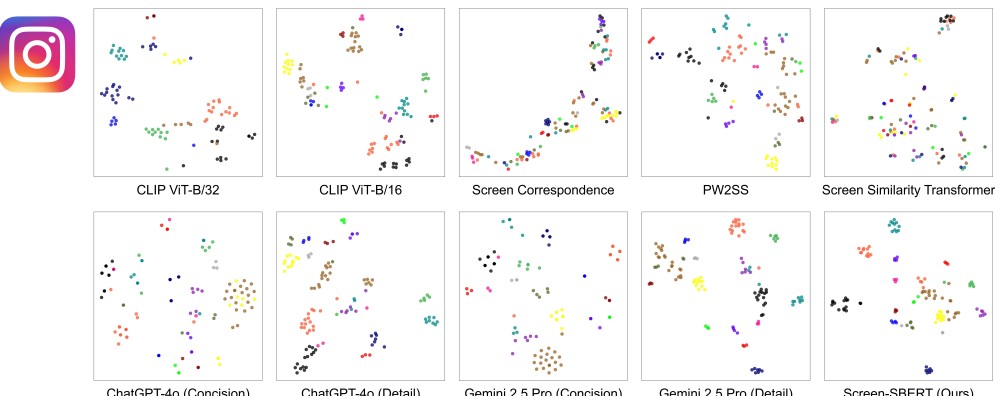

Figure 13: t-SNE plots of the embedding spaces for **Instagram**.

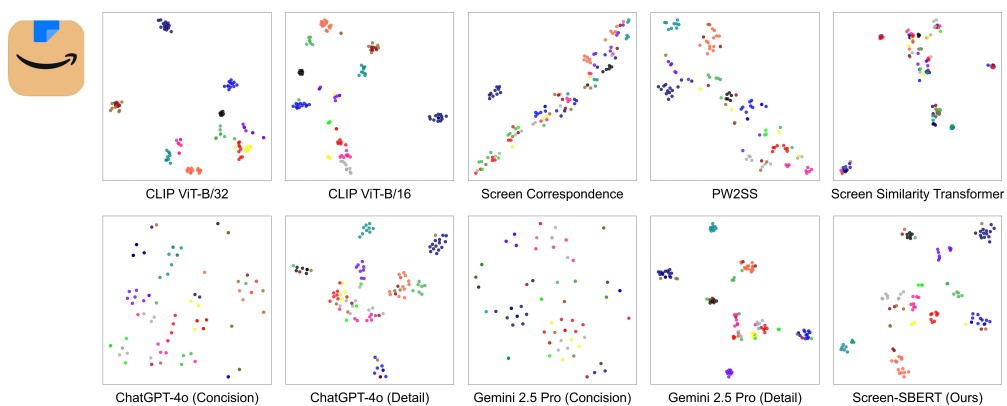

Figure 14: t-SNE plots of the embedding spaces for **Amazon Shopping**.

## K.2 FACEBOOK AND COUPANG

Figures 15 and 16 present t-SNE visualizations of the similarity matrices for Facebook and Coupang. All methods were trained on the remaining four apps—Instagram, X, Amazon Shopping, and Temu. For clarity, only classes with at least five screenshots are displayed, a higher threshold than the three used for Instagram and Amazon Shopping, reflecting the larger dataset size of these apps.

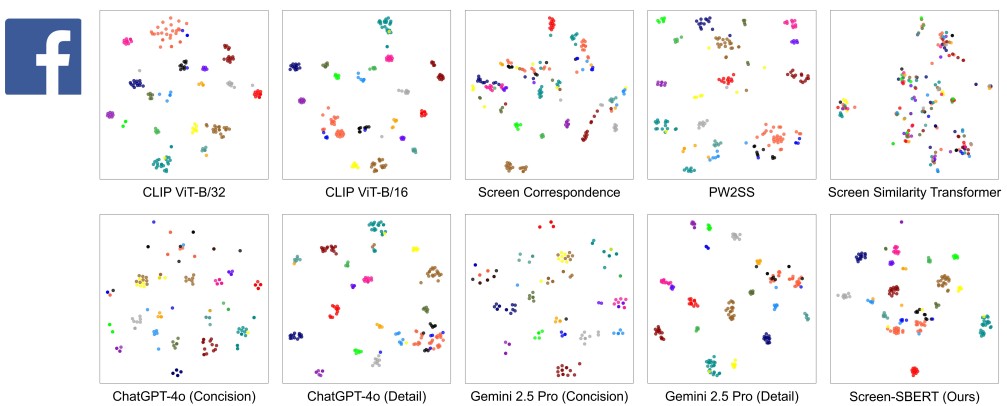

Figure 15: t-SNE plots of the embedding spaces for **Facebook**.

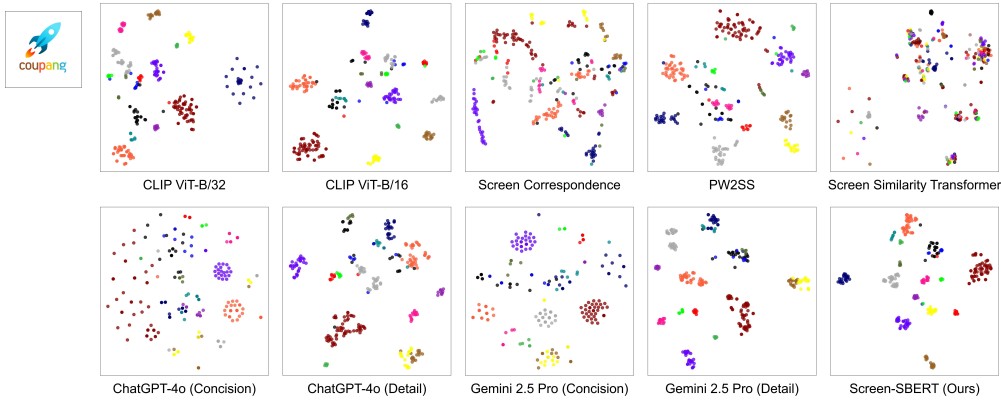

Figure 16: t-SNE plots of the embedding spaces for **Coupang**.

## K.3  X AND TEMU

Figures 17 and 18 present t-SNE visualizations of the similarity matrices for X and Temu. All methods were trained on four apps—Instagram, Facebook, Amazon Shopping, and Coupang. For clarity, only classes with at least five screenshots are displayed.

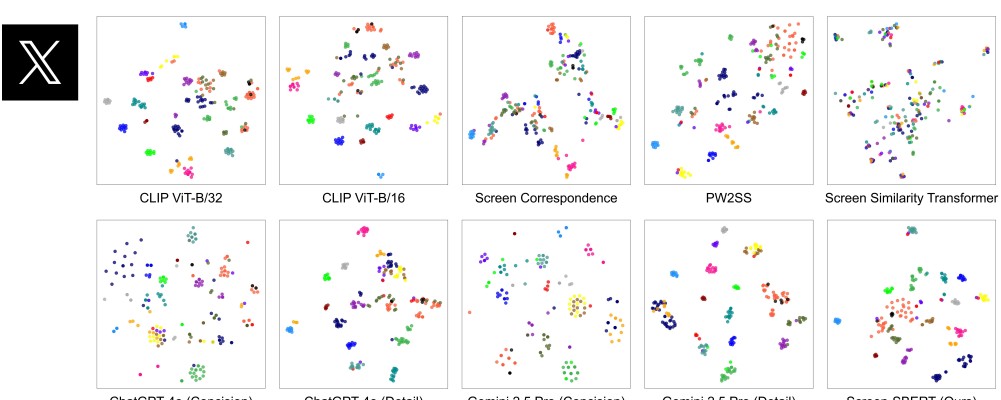

Figure 17: t-SNE plots of the embedding spaces for **X**.

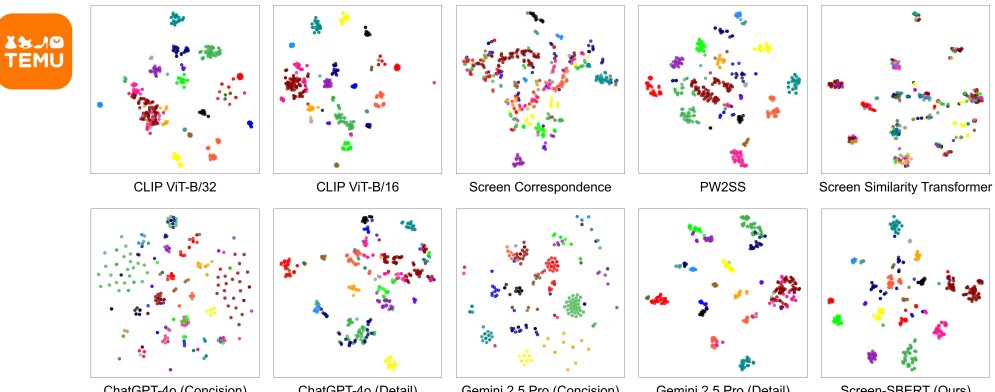

Figure 18: t-SNE plots of the embedding spaces for **Temu**.

## L   SIMULATIONS OF ONE-STEP DECISION WITH SCREEN RETRIEVAL

Given a mobile GUI screenshot and a natural language prompt, we examined whether an LLM—ChatGPT-4o in these examples—could generate a decision appropriate for accomplishing the user's instruction. In this process, we observed that incorporating page-related knowledge into the prompt via screen retrieval enabled the LLM to produce more accurate decisions.

We provided the LLM with a screenshot and a prompt, as illustrated in the following example. We used the coordinates generated by our framework's GUI Parsing Module to overlay bounding boxes with indices on each GUI element in the screenshot. We also included the coordinate information of each element in the natural language prompt. This prompt format was inspired by Mobile-Agent-E (Wang et al., 2025), but to simplify the simulation, we streamlined the prompts to retain only the essential information.

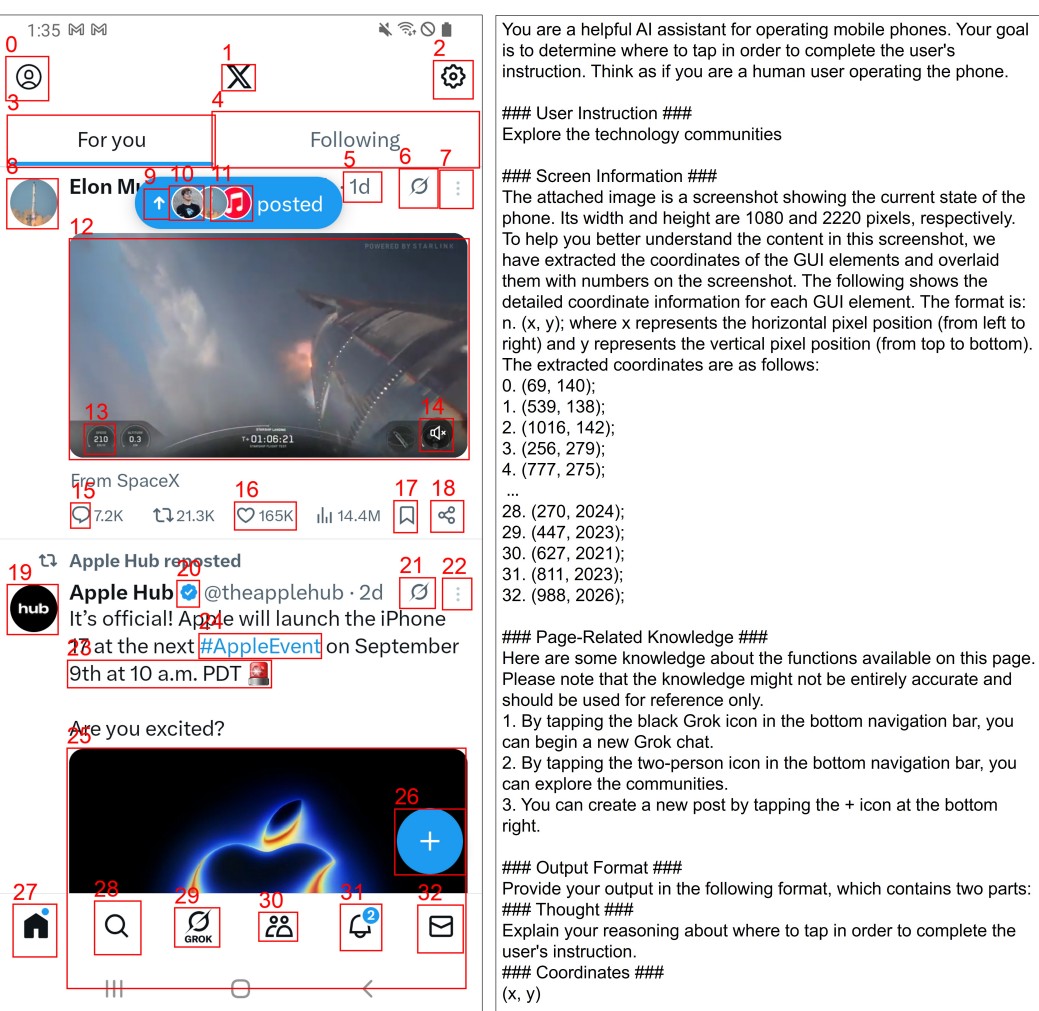

In the **### Page-Related Knowledge ###** section of the prompt, prior knowledge about the identified page is included when the page class of the screenshot is successfully recognized. Without this retrieval step, leveraging such knowledge would be difficult. We assume that this prior knowledge has been constructed in advance, for example through autonomous exploration by an agent or manual authoring by humans. In our examples, we manually specified three pieces of information for each page that the LLM could plausibly confuse with others. Although the prior knowledge itself was manually authored, both the screen retrieval performed by our Screen-SBERT and the decisions made by the LLM were derived directly from actual model outputs.

The following examples are simplified simulations of a one-step decision made by a GUI agent. We believe that refining the perspective presented in this appendix into a more concrete and sophisticated system could further enhance the performance of GUI agents in future work.

## L.1   SIMULATION ON X

Figure 19 illustrates a case on X where the LLM makes an incorrect decision when the prompt does not include Page-Related Knowledge. The user's instruction in this example is "Explore the technology communities." In the X app, fulfilling this instruction requires tapping the two-person icon in the bottom navigation bar. Because the search function does not provide direct access to communities, it can cause the agent to make erroneous subsequent decisions. Lacking such app-specific knowledge, however, the LLM incorrectly inferred that "explore" was related to "search," ultimately resulting in the wrong decision.

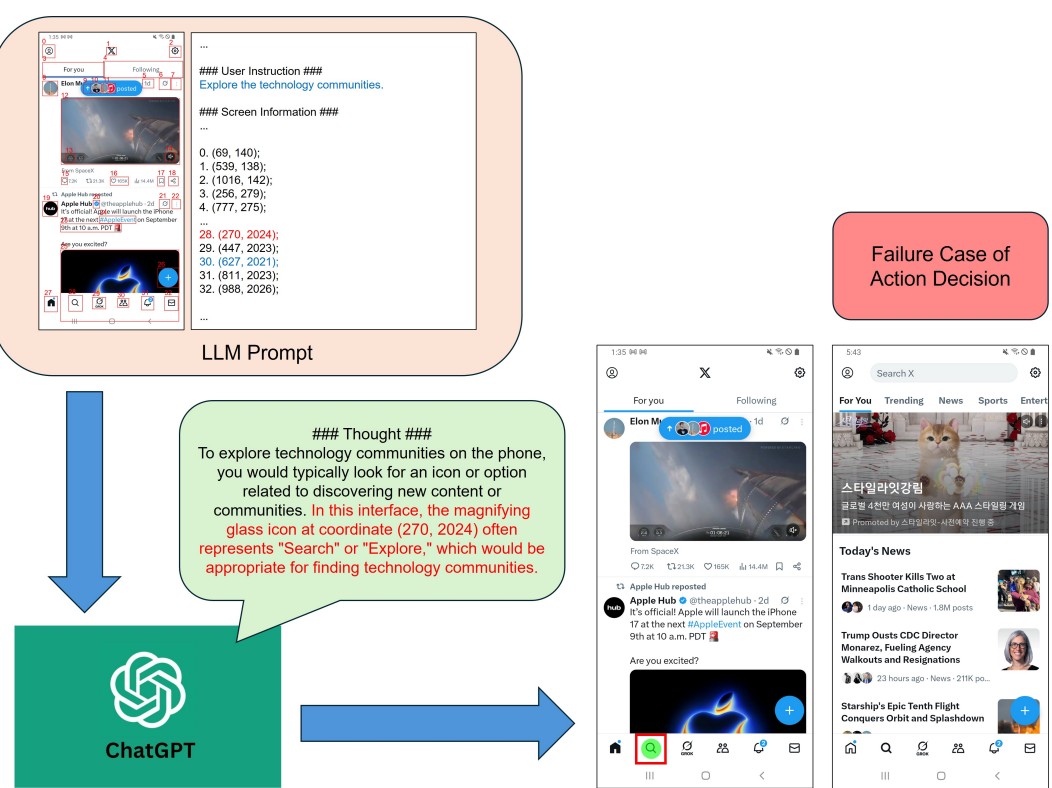

Figure 19: A failure case on X where the LLM generates an incorrect action decision.

Figure 20 presents the actual retrieval result of the screenshot using our Screen-SBERT framework. Since the page class of the retrieved functionally equivalent screen (Home) is already known, we can assume that the input screenshot belongs to the same class and thereby incorporate page-related knowledge into the prompt.

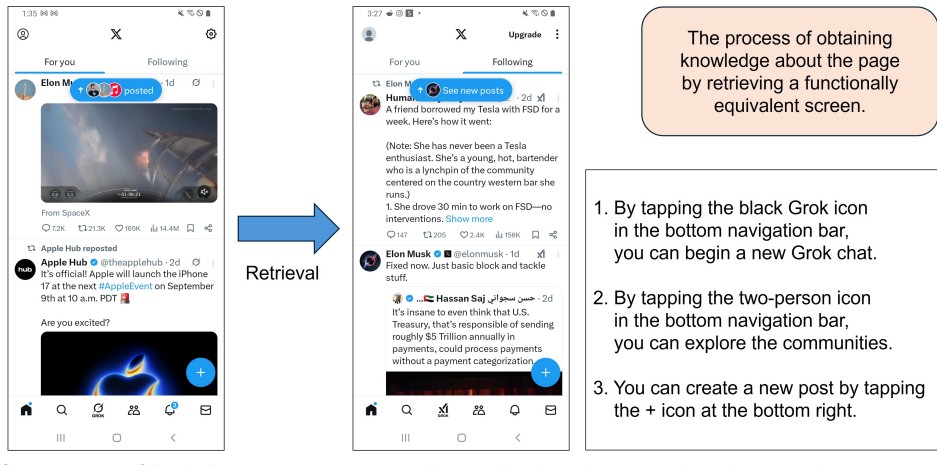

Figure 20: The actual retrieval result of the X screenshot using our Screen-SBERT framework.

Figure 21 illustrates how incorporating the retrieved page-related knowledge into the prompt allows the LLM to make the correct decision.

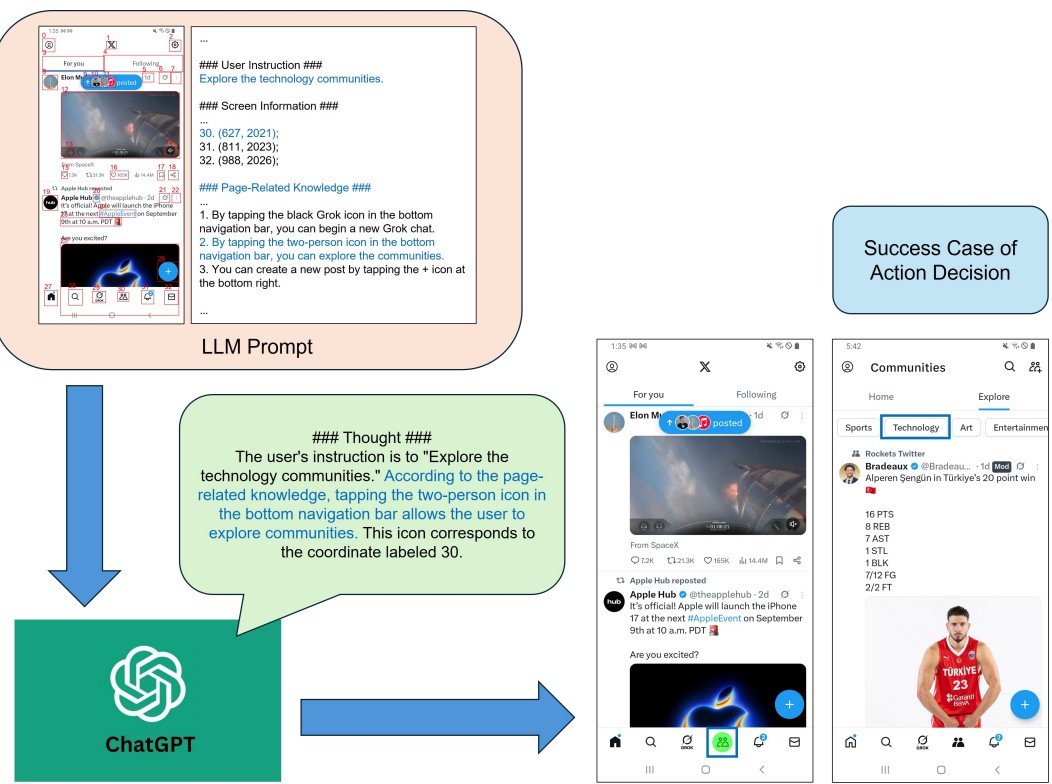

Figure 21: A success case on X where the LLM generates an correct action decision.

## L.2 SIMULATION ON TEMU

Figure 22 illustrates a case on Temu where the LLM makes an incorrect decision when the prompt does not include Page-Related Knowledge. In this example, the user's instruction is "Access the Image Search function to find other products that are visually similar to this one." In the Temu app, fulfilling this instruction requires tapping the product image once. The basic search function chosen by the LLM does not provide the capability to find visually similar products; this feature is revealed only after the image is tapped. Lacking such app-specific knowledge, the LLM incorrectly inferred the action, ultimately leading to the wrong decision.

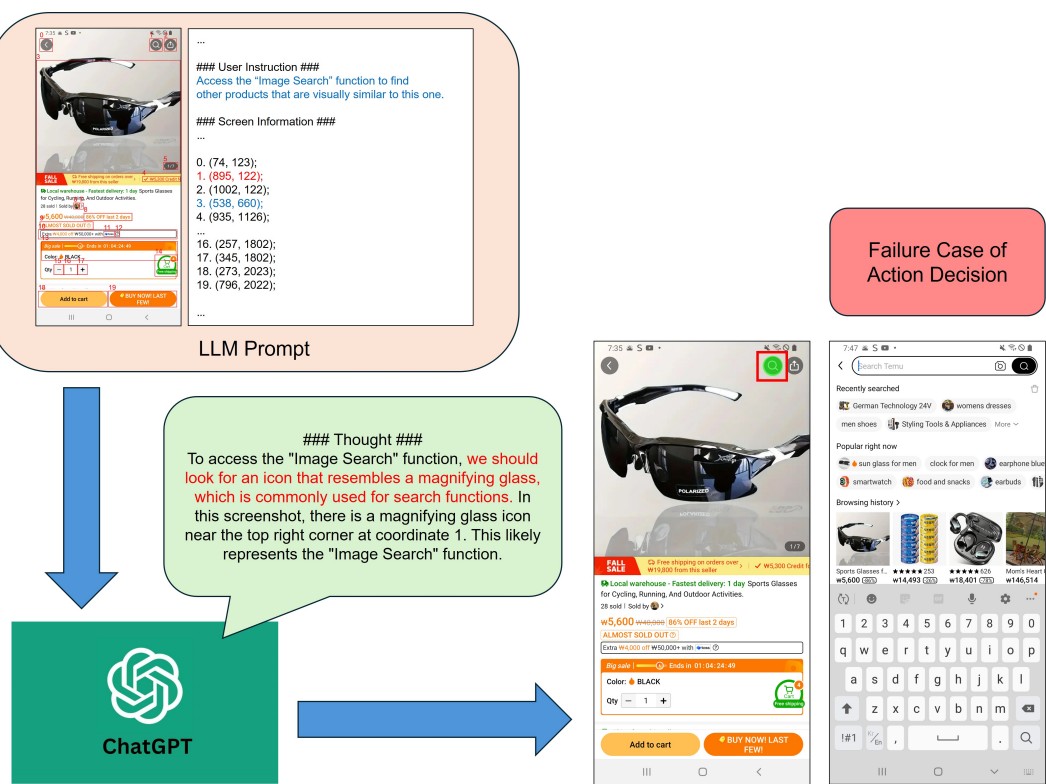

Figure 22: A failure case on Temu where the LLM generates an correct action decision.

Figure 23 presents the actual retrieval result in this simulation. Because the page class of the retrieved functionally equivalent screen (Product Details) is already known, we can assume that the input screenshot belongs to the same class and thereby incorporate page-related knowledge into the prompt.

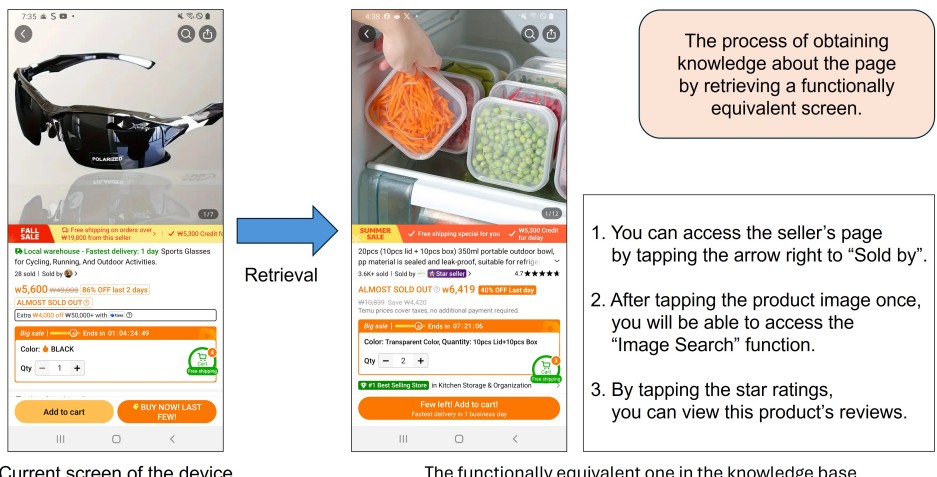

Figure 23: The actual retrieval result of the Temu screenshot using our Screen-SBERT framework.

Figure 24 shows that incorporating page-related knowledge into the prompt enables the LLM to make the correct decision.

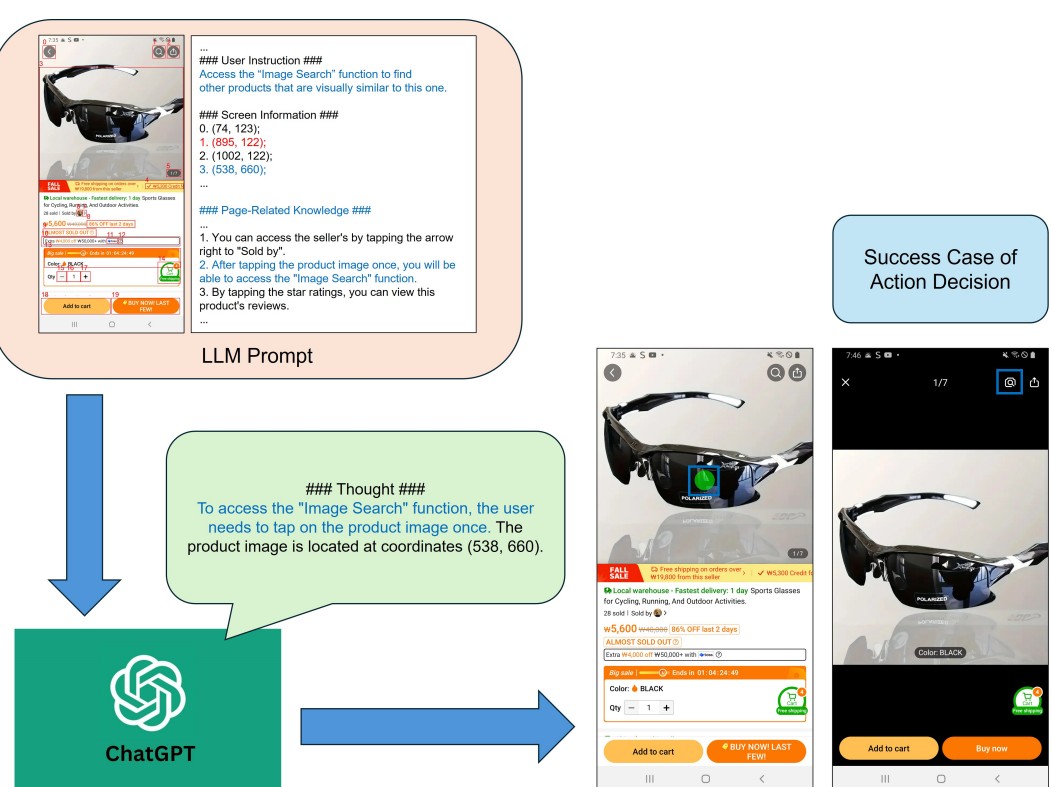

Figure 24: A success case on Temu where the LLM generates an correct action decision.

