# OpenReview forum: "SCREEN-SBERT: EMBEDDING FUNCTIONAL SEMANTICS OF GUI SCREENS TO SUPPORT GUI AGENTS"
_ICLR.cc/2026/Conference — Submitted to ICLR 2026_

### Official Review · Reviewer_mRBN · 2025-10-28

**Soundness:** 4
**Presentation:** 4
**Contribution:** 4
**Rating:** 8
**Confidence:** 4

**Summary:**

The paper addresses the limitation of existing GUI retrieval methods that rely solely on visual similarity, which often leads to mismatches between elements that look alike but serve different functions. To overcome this, the authors use a two-stage retrieval framework:
(1) a screenshot-level retrieval stage that identifies GUI screens sharing similar functional semantics, and
(2) a fine-grained element-level retrieval stage for detailed matching.

The innovation is introduced in the first stage. They introduce Screen-SBERT, a model designed to embed the functional meaning of GUI screenshots using purely visual cues. The framework’s key innovations include:
(1) capturing functional semantics of GUI screens in a vision-only setting, and
(2) employing a contrastive learning approach that supports effective few-shot learning and achieves robust performance even with small datasets.

**Strengths:**

1. Unlike previous works that treat GUI retrieval mainly as a design-assistance task emphasizing visual similarity, this study redefines the task as retrieving functionally equivalent knowledge for augmenting LLM prompt. The introduction of the concept of Functional Equivalence provides a valuable and meaningful new perspective.

2. The authors reimplemented several closed-source models, and their released code closely follows the method reported in the original papers, which is a notable contribution to reproducibility.

3. The paper includes a detailed discussion of design alternatives and decision rationales, making the final framework convincing and well-grounded.

**Weaknesses:**

1. While the paper claims that the proposed method can facilitate knowledge retrieval for augmenting LLM prompts, it lacks experiments or case studies demonstrating this motivation in practice.

2. The work overlooks a potentially important baseline — using a Multimodal LLM (MLLM) to generate functional descriptions for each screenshot and performing retrieval based on these descriptions.

**Questions:**

See weakness.

---

> ### Author Response · Authors · 2025-11-19
>
> # Thanks for Reviewer mRBN #
>
> We sincerely appreciate the time and effort you have devoted to reviewing our paper and for evaluating it favorably. As there were no separate questions apart from the weaknesses, we provide our responses to those weaknesses instead.
>
> ---
>
> # Responses to the Weaknesses #
>
> ### W1) While the paper claims that the proposed method can facilitate knowledge retrieval for augmenting LLM prompts, it lacks experiments or case studies demonstrating this motivation in practice. ###
>
> A1) This is correct. At present, the only evidence we can provide regarding how our method may improve the performance of GUI agents is the set of toy examples presented in Appendix L, and we acknowledge that more systematic evaluations are lacking.
>
> Nevertheless, we believe that the proposed approach has substantial potential when integrated into a full agent system, and we hope that its effectiveness will be explored and validated in future work.
>
> ---
>
> ### W2) The work overlooks a potentially important baseline — using a Multimodal LLM (MLLM) to generate functional descriptions for each screenshot and performing retrieval based on these descriptions. ###
>
> A2) This is a valid point. However, given that there are currently very few cases demonstrating effective retrieval using MLLMs, we believe that an approach whose input prompts are not carefully defined may not serve as a meaningful baseline.
>
> More specifically, an MLLM-based retrieval approach would need to address two challenges: first, designing and engineering an appropriate prompt for the MLLM, and second, determining how to measure similarity and perform nearest-neighbor retrieval based on the descriptions generated by the model. (At this stage, the most intuitive option would likely be to encode the MLLM’s natural-language outputs using Sentence-BERT.)
>
> If an MLLM-based method performs poorly, the primary cause would likely be suboptimal prompt design rather than a limitation of the MLLM itself. We view this issue as suggesting a potentially interesting direction for future research on prompt engineering for GUI-related retrieval.
>
> We may be able to conduct a small-scale retrieval experiment using both ChatGPT-4o and Gemini-2.5 during the rebuttal period. However, given the tight time constraints, we expect that only relatively simple designs can be explored at this stage. We plan to prepare a brief sub-report summarizing these preliminary experiments and will upload it as soon as it is ready. We kindly ask for your understanding regarding the additional time this may require.

---

### Official Review · Reviewer_pJ4Z · 2025-10-31

**Soundness:** 2
**Presentation:** 3
**Contribution:** 2
**Rating:** 4
**Confidence:** 4

**Summary:**

This paper addresses a core challenge in knowledge retrieval for GUI agents: existing methods either over-rely on unreliable structured metadata or merely compare visual appearance , failing to accurately retrieve functionally equivalent GUI screens.
Compared to baselines that rely on unreliable metadata, only match visual appearance, or perform poorly on small datasets due to the use of MLM , Screen-SBERT is a purely vision-based bi-encoder model. It leverages contrastive learning to effectively learn functional semantics from a small dataset and achieves retrieval efficiency far exceeding that of cross-encoder models .
The main limitations are that the annotation of "functional page classes" relies on subjective judgment , all experiments are confined to a small-scale dataset (i.e., few-shot learning) , and computational constraints (a single GPU) prevented comparison against larger VLM baselines.
The paper uses a manually constructed dataset of 1,814 screenshots from six real-world applications . The code and preprocessed modality data are open-sourced on GitHub, but the original screenshots are not released due to privacy concerns .

**Strengths:**

1. The core contribution of this paper is the introduction of the "functional equivalence" concept. This enables the model to retrieve screens that share the same function despite dynamic content (e.g., products, posts) , addressing a fundamental problem where existing purely vision-based methods (like CLIP) are confounded by superficial appearance .

**Weaknesses:**

1. The page class annotation lacks clear, objective criteria. The dataset used for training and evaluation was manually labeled based on the authors' "intuitive judgment" , which introduces "a degree of subjectivity" to the class boundaries . This makes the annotation process difficult to scale and verify.
2. Due to the "lack of a large-scale public dataset" for this task , all experiments in this study fall under "few-shot learning". Therefore, it remains unknown whether the paper's conclusions (e.g., the superiority of contrastive learning over MLM ) would "generalize to large-scale scenarios".
3. Due to "computational constraints (training on a single GPU)" , the authors were unable to conduct fine-tuning experiments against "larger VLM (Vision-Language Models)" that require multiple GPUs. This leaves the SOTA comparison incomplete.

**Questions:**

1. To what extent can the functional semantics learned by Screen-SBERT (particularly its heavy reliance on layout structure ) generalize to application domains with entirely different layout paradigms and interaction logic, such as productivity tools, banking/finance apps, or complex game interfaces?
2. How did you ensure the consistency and reliability of the annotations? Was any form of cross-validation performed to validate the subjective labeling?

---

> ### Author Response · Authors · 2025-11-19
>
> # Thanks for Reviewer pJ4Z #
>
> We sincerely appreciate the time and effort you have dedicated to reviewing our paper. We also thank you for raising the thoughtful and insightful question.
>
> ---
>
> # Responses to the Questions #
>
> ### Q1) To what extent can the functional semantics learned by Screen-SBERT (particularly its heavy reliance on layout structure ) generalize to application domains with entirely different layout paradigms and interaction logic, such as productivity tools, banking/finance apps, or complex game interfaces? ###
>
> A1) To begin with, we cannot reliably determine the generalization capacity of Screen-SBERT to domains outside social media and shopping apps without evaluating the model on data from those domains.
>
> However, we expect that the model will still perform reasonably well even in entirely different domains. This expectation is supported by our experimental findings: in our evaluations, functionally equivalent screens were embedded close to one another even when their layout patterns had never appeared during training. We also note that layout structures vary considerably across apps in today’s mobile ecosystem, even among applications within the same category.
>
> Our underlying view is that the functional semantics of a GUI screenshot are closely tied to its global layout and the contextual relationships among its elements. We believe that humans rely heavily on such structural and relational cues when identifying the functional meaning of a GUI screen, and this intuition guided our annotation of page classes.
>
> Consistent with this perspective, both our approach and several baselines treat the entire screenshot as a “sentence” and individual GUI elements as “words,” reflecting the idea that functional semantics emerge from the global organization of elements rather than from isolated components.
>
> For these reasons, if the model can distinguish functional semantics by leveraging layout differences within the same app, we expect that this capability could extend—at least to some degree—to domains such as finance, productivity tools, or gaming.
>
> ---
>
> ### Q2) How did you ensure the consistency and reliability of the annotations? Was any form of cross-validation performed to validate the subjective labeling? ###
>
> A2) We acknowledge your concern regarding the consistency and potential subjectivity of our functional equivalence annotations. It is true that subjective validity has not yet been formally verified. Therefore, during the rebuttal period, we plan to partially validate these annotations.
>
> We have developed a simple verification tool that allows external participants to review the page-class boundaries we defined (i.e., the grouping criteria for screenshots) and indicate whether they agree with them. We have also recruited several participants to conduct this verification task. They are expected to finish reviewing the dataset by November 23, after which we will compute the level of agreement between their judgments and our subjective annotations and include the results in an additional appendix.
>
> More concretely, each participant will examine every screenshot in the dataset and indicate whether they agree with the assigned page class. We will then compute how many screenshots they disagree with, relative to the total number of screenshots, in order to quantify the difference from our original annotations.
>
> Most of the participants are our personal acquaintances who do not have expertise in machine learning theory. (We ask for your understanding that, due to the privacy-sensitive nature of the screenshots in the dataset, the pool of reviewers had to be limited to personal acquaintances.) Because they have limited exposure to machine learning, we expect them to approach the task without technical preconceptions. In other words, we expect them to review the functional-equivalence annotations from a more general, non–ML-influenced perspective.
>
> We will collect the verification results by November 23, analyze them, and include the findings in an updated appendix. We will upload the revised paper once this process is completed. We appreciate your patience as we work to supplement the annotation validation.

---

### Official Review · Reviewer_6YbJ · 2025-11-01

**Soundness:** 2
**Presentation:** 3
**Contribution:** 3
**Rating:** 6
**Confidence:** 3

**Summary:**

This paper introduces Screen-SBERT, a vision-based framework for learning functional embeddings of GUI screenshots in mobile apps. The core motivation is to support GUI agents by enabling retrieval of functionally equivalent screens (e.g., “Home” or “Product Details”) even when their visual content differs. Screen-SBERT uses a bi-encoder architecture inspired by Sentence-BERT and trains with contrastive learning to embed screenshots based on functional semantics rather than pixel similarity. It does not rely on metadata like view hierarchies, which are often unavailable or outdated.

The framework consists of:
1. A GUI Parsing Module that extracts multimodal features (vision, text, coordinates, functional type) for each GUI element.
2. A GUI Embedding Module that converts these features into unified token embeddings.
3. A Transformer encoder with 2D relative positional bias to model spatial layout.
4. A contrastive learning objective (InfoNCE) to learn screenshot-level embeddings.

Experiments are conducted on a manually curated dataset of 1,814 screenshots from 6 apps (Instagram, Facebook, X, Amazon, Coupang, Temu), with out-of-domain evaluation. Screen-SBERT outperforms baselines (CLIP, Screen Correspondence, PW2SS, Screen Similarity Transformer) in retrieval accuracy and efficiency.

**Strengths:**

1. It is the first work to formally define functional equivalence and functional page class for GUI screenshots.
2. The proposed method outperforms baselines in Macro F1 and Top-1 accuracy across multiple OOD settings.
3. This paper thoroughly analyzes the impact of modalities, positional encodings, and training objectives.

**Weaknesses:**

1. The dataset is somewhat small. Only 1,814 screenshots from 6 apps; limited generalizability to larger or more diverse apps.
2. Only CLIP-Base is used; no comparison with larger VLMs like CLIP-Large or SigLIP due to GPU constraints.
3. Functional equivalence is labeled manually by authors; no inter-annotator agreement reported.

**Questions:**

1. Have you considered evaluating cross-app retrieval (e.g., retrieve “Home” screen from Temu using Instagram’s “Home” as query)? This would test whether the model learns general functional semantics rather than app-specific layout patterns.
2. How consistent are your functional equivalence annotations? Were multiple annotators involved? If not, how do you ensure that the model is not simply memorizing your subjective grouping criteria?
3. Your rule-based classifier for GUI element types is brittle. Have you tried fine-tuning a small classifier on a few hundred labeled examples instead of relying on captioning? This could be a lightweight improvement.

---

> ### Author Response · Authors · 2025-11-19
>
> # Thanks for Reviewer 6YbJ #
>
> We sincerely appreciate the time and effort you have devoted to reviewing our paper and for evaluating it favorably. We are also deeply grateful for the thoughtful questions you have provided.
>
> ---
>
> # Responses to the Questions #
>
> ### Q1) Have you considered evaluating cross-app retrieval (e.g., retrieve “Home” screen from Temu using Instagram’s “Home” as query)? This would test whether the model learns general functional semantics rather than app-specific layout patterns. ###
>
> A1) Cross-app retrieval was not considered within the scope of our work.
>
> When designing our framework, we placed strong emphasis on encoding the global layout and the positional relationships among GUI elements. Our intuition is that humans typically rely on such layout structure and contextual relationships when identifying the functional semantics of a GUI screenshot. This same intuition guided our annotation of page classes in the dataset.
>
> In particular, the paradigm adopted in this work—as well as in several baselines—treats the entire screenshot as a “sentence” and each GUI element as a “word.” This perspective reflects our view that humans primarily perceive a GUI screenshot by interpreting the global layout and relational organization of its elements, rather than by analyzing individual components in isolation.
>
> From this viewpoint, the idea that certain GUI elements and their contextual arrangement constitute a “Home” page becomes clearer when contrasted with other pages within the same app. In other words, we regard the “Home” page class not as an absolute concept, but as one that is defined only in relation to other page classes within the same application.
>
> The “Home” screens of different apps are often developed with substantially different layout patterns. Under this intuition, our framework would therefore have difficulty identifying an absolute, layout-independent notion of functional semantics in a cross-app setting.
>
> As you suggested, if a model were able to learn layout-independent functional semantics and perform cross-app retrieval, we believe this would go beyond what is achievable through human perceptual cues alone. For this reason, we consider it an especially interesting and valuable direction for future work. Thank you for raising this perspective.
>
> ---
>
> ### Q2) How consistent are your functional equivalence annotations? Were multiple annotators involved? If not, how do you ensure that the model is not simply memorizing your subjective grouping criteria? ###
>
> A2) We acknowledge your concern regarding the consistency and potential subjectivity of our functional equivalence annotations. At the time of writing the paper, these annotations were created by a single annotator, and their subjective validity has not yet been formally verified. Therefore, during the rebuttal period, we plan to partially validate these annotations.
>
> We have developed a simple verification tool that allows external participants to review the page-class boundaries we defined (i.e., the grouping criteria for screenshots) and indicate whether they agree with them. We have also recruited several participants to conduct this verification task. They are expected to finish reviewing the dataset by November 23, after which we will compute the level of agreement between their judgments and our subjective annotations and include the results in an additional appendix.
>
> More concretely, each participant will examine every screenshot in the dataset and indicate whether they agree with the assigned page class. We will then compute how many screenshots they disagree with, relative to the total number of screenshots, in order to quantify the difference from our original annotations.
>
> Most of the participants are our personal acquaintances who do not have expertise in machine learning theory. (We ask for your understanding that, due to the privacy-sensitive nature of the screenshots in the dataset, the pool of reviewers had to be limited to personal acquaintances.) Because they have limited exposure to machine learning, we expect them to approach the task without technical preconceptions. In other words, we expect them to review the functional-equivalence annotations from a more general, non–ML-influenced perspective.
>
> We will collect the verification results by November 23, analyze them, and include the findings in an updated appendix. We will upload the revised paper once this process is completed. We appreciate your patience as we work to supplement the annotation validation.

---

> > ### Author Response · Authors · 2025-11-19
> >
> > ### Q3) Your rule-based classifier for GUI element types is brittle. Have you tried fine-tuning a small classifier on a few hundred labeled examples instead of relying on captioning? This could be a lightweight improvement. ###
> >
> > A3) As noted in Appendix F.2 of the paper, we agree that using a better-trained classifier would further improve the overall performance of the framework. However, training such a classifier requires appropriately labeled GUI image data.
> >
> > Although a model can be fine-tuned with a small amount of data (on the order of a few hundred images), as mentioned in the question, in practice we find that GUI elements serving the same function often exhibit highly diverse visual appearances across different apps, while visually similar elements may perform entirely different functions. Owing to these characteristics, we expect that such a small dataset would be insufficient to achieve meaningful generalization.
> >
> > In fact, we are conducting a sub-project aimed at training a classifier for individual GUI elements. However, we have found it difficult to obtain satisfactory generalization with limited data, and our current conclusion is that, in the present industrial environment, substantially larger and more diverse datasets—covering as many apps as possible—are required.
> >
> > The Florence-2 model from OmniParser, which we used in this work, was a practical choice because it covers a relatively broad range of GUI elements compared to other publicly available models and offers a certain degree of generality. However, the captions produced by Florence-2 do not consistently contain predictable keyword patterns, which limits its reliability for stable classification. For this reason, the Florence-2–based rule-driven classification approach adopted in this paper should be regarded only as a temporary workaround.
> >
> > Consequently, the results presented in this work should be interpreted as demonstrating the level of performance achievable when a heuristic, rule-based classifier is used in scenarios where training a high-quality classifier is not feasible.

---

### Official Review · Reviewer_ixGd · 2025-11-02

**Soundness:** 2
**Presentation:** 3
**Contribution:** 2
**Rating:** 4
**Confidence:** 4

**Summary:**

This paper proposed Screen-SBERT, a vision-based framework for embedding and retrieving the functional semantics of GUI screenshots to support GUI agents in mobile apps. Unlike prior work that relies on structured metadata, it uses only visual information to identify functionally equivalent screens. The method also employs contrastive learning for learning GUI embedding. Evaluation results show that Screen-SBERT outperforms baselines in retrieving functionally similar screens.

**Strengths:**

+) This paper proposed a pure-vision solution to GUI embedding which not requires view hierarchy

+) Interesting ablation study to reveal the importance of each modality

**Weaknesses:**

-) This paper lacks comparison to methods that use view hierarchy when they are available, to better understand the gap

-) The evaluation is limited to a relatively small, manually constructed dataset

**Questions:**

a) How does the approach compare to methods that use structured metadata, both in terms of accuracy and computational efficiency?

b) How does Screen-SBERT handle screens with dynamic or context-dependent elements that may not be visually distinguishable but differ functionally?

---

> ### Author Response · Authors · 2025-11-19
>
> # Thanks for Reviewer ixGd #
>
> We sincerely appreciate the time and effort you have dedicated to reviewing our paper. In the following, we provide our responses to your comments and questions.
>
> ---
>
> # Responses to the Questions #
>
> ### Q1) How does the approach compare to methods that use structured metadata, both in terms of accuracy and computational efficiency? ###
>
> A1) First, we provide a brief clarification of how existing agent systems typically utilize structured metadata. For example, in an “AppAgent” system, the Android XML layout is extracted, the unique resource ID of each GUI element is obtained, and functional knowledge is stored in text files named after these IDs (e.g., {resource_id}.txt). During testing, the system again extracts the XML layout, identifies the resource IDs, and simply reads the corresponding files to incorporate the stored knowledge.
>
> This design is indeed highly advantageous in terms of accuracy and computational efficiency, since it directly uses stable resource IDs as keys and allows immediate file look-up. However, it has a fundamental limitation in that it is not platform-independent. In particular, this mechanism is not applicable to iOS, and it is well known that many real-world industrial environments do not allow reliable extraction of XML layouts. Recent studies have highlighted these constraints and argue that the field should gradually reduce reliance on metadata and move toward vision-only approaches.
>
> We share this perspective. Our work is positioned as an initial exploration of what a purely vision-based method can achieve. Rather than aiming to directly compete with metadata-based approaches in efficiency, the goal of this paper is to offer ideas and perspectives that may contribute toward future GUI understanding systems that can operate without metadata dependence. In other words, this work serves as an early exploratory step toward replacing non-ML, metadata-driven mechanisms with machine-learning-based ones.
>
> ---
>
> ### Q2) How does Screen-SBERT handle screens with dynamic or context-dependent elements that may not be visually distinguishable but differ functionally? ###
>
> A2) The scenario described in the question can be interpreted in several ways, so we address the most likely cases based on our understanding.
>
> First, if the question refers to situations where a small popup or overlay appears on an otherwise identical page and restricts certain interactions, such screens are annotated as separate page classes in our dataset.
>
> In contrast, for cases where only a local button state changes—such as Follow/Unfollow, adding/removing favorites, or toggling a bookmark—while the overall page structure and functional purpose remain the same, we treat these screens as belonging to the same functional page class. This is because we consider the global functional context of the screenshot to remain largely intact.
>
> If the intended scenario corresponds to a different type of “visually similar but functionally different” screen, we would appreciate a concrete example so that we can provide a more precise and targeted response.
>
> ---
>
> # Reference #
>
> [1] AppAgent: https://github.com/TencentQQGYLab/AppAgent.git

---

### Author Response · Authors · 2025-11-26

# Paper Revision \#1 #

We sincerely thank all reviewers for their patience in waiting for the revision. We have uploaded the revised version of the paper.

To partially assess the consistency and degree of subjectivity in our annotations, we developed a simple verification tool and recruited nine external participants to perform the validation task. The results of this assessment have been added to Appendix C.

To connect the main paper with the new appendix, we added an additional paragraph to Section 10 (Limitations and Future Works), which required the use of an extra page, bringing the total length to ten pages. Since an additional page was already necessary, we also took this opportunity to move the framework overview figure from the appendix into the main body of the paper to facilitate a clearer understanding of Section 5.

Apart from the changes described above, no other modifications have been made. We thank the reviewers once again for their time and thoughtful feedback.

---

### Author Response · Authors · 2025-11-30

# Paper Revision #2 #

Reviewer mRBN noted that our initial submission overlooked an important baseline: using a Multimodal LLM (MLLM) to generate functional descriptions for each screenshot and performing retrieval based on those descriptions. In response, we conducted additional experiments using two MLLMs—ChatGPT-4o and Gemini 2.5 Pro—and incorporated the full results into this revised version. To reflect these new experiments, we expanded the text in Section 7 of the main paper, added a new Appendix I, and updated the figures in Appendices J and K accordingly. No other modifications have been made beyond these updates.

We refer the reviewer to the following discussion in the main body of the revised paper for the conclusions drawn from these additional experiments:

"Regarding the results of the MLLM-based approaches, we observed a notable improvement when encoding detailed descriptions rather than concise outputs. This is likely because compressing a screenshot description into a short phrase introduces substantial information loss.

However, even with the detailed descriptions, the overall performance remained comparable to that of the other baselines and fell short of our framework. One plausible explanation is that the natural-language outputs of the MLLMs had to be re-encoded by an mGTE model, during which additional information may have been lost. If embedding vectors could be obtained directly from these models without passing through natural language generation, the retrieval performance might have been higher. Nonetheless, such a setup would be conceptually similar to scaling up our framework and the other baselines with larger training data."

We refined the natural language prompt provided to the MLLMs through multiple rounds of trial and error, and we included in the paper the results obtained using the version of the prompt that achieved the best retrieval performance during the rebuttal period.

However, as we noted in our initial response to reviewer mRBN, the prompt we used may still be suboptimal. Although our framework continued to outperform the MLLM-based approach in the experiments conducted during the rebuttal period, this gap may stem not from inherent limitations of the MLLMs themselves but from the possibility that the prompt was not fully optimized.

Thus, we view MLLM-based retrieval—and the corresponding prompt engineering required for it—as an interesting direction for future research. We sincerely thank Reviewer mRBN for motivating this revision.

---

### Meta-Review · Area_Chair_1a5M · 2026-01-04

**Summary:**

This paper proposes an approach for creating a feature representation for GUI screens that does not rely on meta-data like view hierarchies.  Generally speaking, the authors aim to detect GUI elements, compute some token-level feature, and pool them into a single view feature.  In this regard, the setup is extremely similar to the "bottom-up" features long used for natural images (see [A] and a very long and diverse line of subsequent work).  Reviewers questioned both the problem setup (i.e., not using view-hierarchies) as well as the experimental setup.  With regards to the latter, these included the scale of the models run, the dataset collected, its ability to generalize, as well as the accuracy of that collected dataset's labels.  The datasets labels were a particular concern, and the attempts at addressing this in the rebuttal were not sufficient (more on that later).  This could be ignored if the authors showed that their work, in fact, provided effective "support <of> GUI Agents," but despite the inclusion of this in the paper's title, I couldn't find work on downstream tasks demonstrating the benefit of the proposed feature representation.  Instead, the authors only evaluated screen retrieval, which could theoretically help GUI agents, but is not guaranteed (see Table 3 of [B]).  As such, even arguing this paper as a application-specific version of [A] lacks sufficient evidence it is useful in its current form.

[A] Bottom-Up and Top-Down Attention for Image Captioning and Visual Question Answering. CVPR 2018

[B] A Dataset for Interactive Vision-Language Navigation with Unknown Command Feasibility. ECCV, 2022

**Reviewer Concerns:**

Some concerns with regards to a few newer models and ablations were successfully addressed in the rebuttal.  However, the authors did not demonstrate that this approach could generalize to unseen apps.  Even newer versions of the same app would be useful here.  Instead, they more or less argued why it would be a challenge for their approach.  This highlights a weakness of their approach, which I am sure the authors would argue could be eliminated in subsequent work on their task.  This *might* end up being true, but is uncertain and subsequent work could also find it is simply a fundamental flaw in their approach.

The second issue I will point to is the small scale of their data and annotations.  These are fundamental limitations that were not contested.  Further, the experiments provided to validate the quality of their annotations, which was raised by multiple reviewers, was insufficient.  Specifically, the authors essentially asked annotators to indicate whether one of their groupings was good.  However, this leaves open the possibility that, given the complexity of this task (i.e., reasoning about multiple screenshots), the annotators are simply a poor judge. I.e., that agreement for even very bad groupings may be high.  This is further complicated by the fact that key elements are not properly described. For example, the authors say they used a 5 point Likert scale, but did not define what they are.  It could be that they meant 0 means strongly disagree and 5 being strongly agree, but it could be that it was all meant to be varying levels of agreement (i.e., that only 0 indicates a completely failed grouping). This would have *very* different interpretations of the results, and reading the authors paper I could not find an answer to this point.  A stronger experiment would have used A/B testing at the very least.

**Reviewer Scores:**

Given the importance of the unresolved concerns, I do not see it likely any changes in reviewer score would have resulted in better scores, and a good argument remains that they might have been lowered in the end.

---

### Decision · Program_Chairs · 2026-01-26

Reject